# The Use of the Thiol-Ene Addition Click Reaction in the Chemistry of Organosilicon Compounds: An Alternative or a Supplement to the Classical Hydrosilylation?

**DOI:** 10.3390/polym14153079

**Published:** 2022-07-29

**Authors:** Ilya Krizhanovskiy, Maxim Temnikov, Yuriy Kononevich, Anton Anisimov, Fedor Drozdov, Aziz Muzafarov

**Affiliations:** 1A. N. Nesmeyanov Institute of Organoelement Compounds, Russian Academy of Sciences, Moscow 119334, Russia; ilya.krizhanovskiy@mail.ru (I.K.); temnikov88@gmail.com (M.T.); yuscience@mail.ru (Y.K.); 2Enikolopov Institute of Synthetic Polymeric Materials, Russian Academy of Sciences, Moscow 117393, Russia; drozdov@ispm.ru

**Keywords:** thiol-ene, click reaction, organosilicon compounds, hydrosilylation

## Abstract

This review presents the main achievements in the use of the thiol-ene reaction in the chemistry of silicones. Works are considered, starting from monomers and ending with materials.The main advantages and disadvantages of this reaction are demonstrated using various examples. A critical analysis of the use of this reaction is made in comparison with the hydrosilylation reaction.

## 1. Introduction

Organosilicon polymers are widely used in nearly all fields of human activity. Such a broad application range is due, in particular, to the diversity of the chemical composition and architectural forms of these compounds.

The hydrosilylation reaction is among the most versatile tools for creating Si-C bonds. This reaction occurs with participation of metal complex catalysts, most often those of the platinum group, on substrates containing double or triple bonds and organosilicon compounds with a functional Si-H group, is simple to implement, and features high commercial availability and versatility [1,2,3,4]. Publications on the hydrosilylation of unsaturated compounds first appeared in the second half of the past century, but this method acquired practical value for the synthesis of organosilicon products upon introduction of modern platinum catalysts, such as the Speier and Karstedt catalysts [5,6,7]. Hydrosilylation reactions are commonly used in modern industry to produce functional silanes, silicone coatings, and rubbers. Despite the undoubted success of this reaction that utilizes platinum group catalysts, it has a significant drawback, namely, the catalysts used are expensive. Therefore, numerous works deal with the development of commercially more accessible catalysts based on metals of the iron triad [8]. At the current development stage of organosilicon chemistry, the majority of carbosilane systems with various architectures, from linear polymers to branched structures such as comb polymers, dendritic polymers, dendrimers and similar structures, are synthesized using the hydrosilylation reaction.

Along with hydrosilylation, the thiol-ene polyaddition reaction is popular, the latter becoming one of the main methods for the synthesis and functionalization in the chemistry of polymers and individual compounds, in particular, organosilicon ones. Similar to hydrosilylation, this reaction involves the addition of a mercapto group to an unsaturated bond to give α- and/or β-addition products, depending on the process conditions. Reactions of unsaturated compounds with thiols and their analogs have been known for a long time and have been studied rather thoroughly. In fact, the vulcanization of natural (poly-cis-isoprene) rubbers with sulfur patented in the middle of the 19th century by Charles Goodyear was the beginning of thiol-ene chemistry. Two different reaction mechanisms, namely, the ionic and radical ones, were described in sufficient detail back in the 1970s [9]. However, its active use in various fields of chemistry began only over the last twenty years [10], and the interest in this process is steadily increasing. It is worthy of note that the vast majority of works on thiol-ene addition, including the synthesis of silicones, apply the radical process mechanism that is most often initiated photochemically [11,12,13]. This fact can be explained by a number of advantages of this approach that will be described below. 

The existence of commercially available thiol-containing organic and organosilicon derivatives allows a broad range of synthetic approaches to be used for synthesizing compounds with various molecular architectures. In this review, we summarized the literature data on the synthesis and application of organosilicon compounds and materials based thereon obtained by the thiol-ene addition reaction. The main advantages and drawbacks of this reaction in comparison with the “classical” hydrosilylation reaction are identified.

### 1.1. Mechanism of the Process

The hydrothiolation reaction can occur by one of the following mechanisms:ionic mechanism;radical mechanism;

The **ionic mechanism** is typically implemented by one of the two variants [14]:nucleophilic addition is efficient for compounds with an electron-acceptor group at a conjugated double C=C bond. It is performed under basic catalysis conditions and a β-addition product predominates (Michael reaction);electrophilic addition is efficient for compounds with an electron-donor group at a double C=C bond. It is performed under acid catalysis conditions and an α-addition product predominates.

The ionic mechanism requires a thorough adjustment of reaction conditions and is nontolerant to functional groups and chemical bonds prone to cleavage under ionic reaction conditions, which is especially important in the chemistry of organosilicon compounds that often contain Si-O-Si, Si-O-C, and Si-H bonds. Uncontrolled cleavage of these bonds results in undesirable side products (often high-molecular-weight ones). Thus, thiol-ene addition reactions that occur by this mechanism are not widely used in organosilicon chemistry.

Nevertheless, examples of ion-catalyzed hydrothiolation of organosilicon compounds can be found in modern literature. For example, the hydrothiolation of ethynyl-, allyl-, and 2-methylallyl-trialkylsilanes in the presence of scandium triflate (Lewis acid) was studied [15]. It was shown that all the hydrothiolation reactions, both of allyl- and (2-methyl)-allyl-silanes, resulted in addition according to Markovnikov’s rule (see Figure 1).

In the case of hydrothiolation of ethynyl-containing silanes, the reaction regioselectivity was found to depend on the volume of substituents at the silicon atom (see Figure 2).

This work clearly shows that steric factors can play a significant role in the thiolation of organosilicon compounds. It should be noted that only silanes with alkyl substituents at the silicon atom were used in this work.

An example of nucleophilic hydrothiolation in the chemistry of organosilicon compounds that somewhat contradicts the above example can be found in another study [16] where this reaction was used to conveniently obtain carbosilane dendrimers with Si-O-Si bridges using an electron-deficient acrylate olefin (see Figure 3). They succeeded in avoiding the cleavage of the Si-O-Si bond, though a strong organic base, DBU (pK_a_ (DBU) = 24.3), was used.

The next dendrimer generation was obtained by radical thiol-ene addition. The **radical thiol-ene** and thiol-yne addition are much more popular and are referred to as “click reactions”. This type of addition has a number of advantages, namely [17]:low activation energy;weak effect of atmospheric oxygen (mainly but not always);a wide selection of initiation methods depending on the substrate, including those not requiring an initiator;high regioselectivity, i.e., preferential formation of a product of addition to the terminal carbon atom (in the case of a terminal double bond);high reaction rate.

The reactions that follow the radical mechanism are typically performed by the following variants:thermal initiation;with addition of an initiator of radical processes;without a thermal initiator;photoinitiation;with addition of a photoinitiator;without a photoinitiator.

These techniques were covered in detail in another review [18].

The need to use a radical initiator or other activators is mainly determined by the specific structure of the reacting compounds, but sometimes also by the external process conditions [17]. 

Figure 4 shows a scheme of the hydrothiolation reaction mechanism, i.e., thiol addition to an alkene to give a thioether. At the first stage, the initiator withdraws a hydrogen atom from the thiol to give a thiol radical (initiation stage (1)), which then adds to the olefin (2). This process results in an intermediate with a radical center at the carbon atom. Further, this intermediate follows the chain transfer pathway (3) resulting in the main product and generation of another thiol radical. However, it can also follow the macromolecular chain growth (polymerization) pathway, which again results in a carboradical [19]. The ratio of these processes is determined by the nature of the olefinic substrate. For example, only chain transfer is observed in the thiolation of norbornene, whereas these reactions compete in the case of acrylates [20].

Thus, the desired product is formed at the chain transfer step. It is well known that this is the limiting stage in many hydrothiolation reactions. Hence, it follows quite naturally that the better chain transfer agent the reacting thiol is, the better the reaction occurs [21]. For example, thiol radicals with electron-acceptor substituents that can form resonance-stabilized radicals (such as phenylthiol and thioacetic acid) are more reactive than thiols with electron-donating substituents (e.g., butylthiol) [19]. This increases the stability of the resulting radical and hence ensures its longer life time and hence its higher concentration in the mixture, which in turn increases the reaction rate. Rather recently, the first theoretical study of the dependence of the hydrothiolation kinetics on the functionality of thiols was performed using DFT calculations [22]. Comparison of phenylthiols with different substituents at the para-position (MeO, H, Cl, and NO_2_) led to the conclusion that thiols with electron-accepting substituents provide a lower activation energy. 

Numerous studies of the hydrothiolation reaction have shown that terminal olefins with electron-donating substituents at the double bond are more reactive [23]. Moreover, early studies on the hydrothiolation of various functional olefins showed that the presence of a heteroatom at the terminal double bond has a strong activating effect by itself that directs the reaction to the beta-position [21]. This is confirmed both by the first studies that have already become classical [24] and by more recent studies performed using computer simulation methods [20]. 

In yet another work [20], the kinetics of hydrothiolation of various olefins with methyl mercaptan was studied and it was shown that the reactivity of an olefin decreases in the case of compounds capable of forming a resonance-stabilized carbo-centered radical (see Figure 5). 

Since silicon is a more electropositive element than carbon (the relative electronegativity according to Pauling is 1.9 vs. 2.5, respectively), vinyl groups at the silicon atom are characterized by a shift of electron density to the terminal carbon atom. It is known that the thiyl radical is an electrophile [24]; therefore, its addition occurs rather quickly and selectively to the beta-position. Vinylsilanes comprising strong electron-accepting substituents, which react much more slowly, are an exception. A work may be cited where this conclusion was made by comparing the reactivity of trichloro-, trialkoxy-, and trimethyl-vinylsilanes in the reaction with ethanedithiol [25].

It is not always possible to get rid of an alpha-adduct completely. However, in the hydrothiolation of organosilicon compounds, the selectivity with respect to an alpha-adduct rarely exceeds 5%. The vast majority of authors report extremely high rates, yields, and selectivities of radical reactions of thiols with a vinyl group at the silicon atom. For example, hydrothiolation of tetravinylsilane with various functional thiols was performed [26]. In all the cases, the content of the alpha-adduct did not exceed 5%. This result can be explained by both chemical factors described above and steric factors. Nearly all organosilicon compounds have rather bulky substituents at the silicon atom that make it difficult for the sulfur atom to approach the alpha position.

Thus, the very nature of this reaction favors its very convenient and flexible application in organosilicon chemistry. This is confirmed by experimental data: a review [27] gives numerous examples of works on the thiolation of vinyl-containing silanes, siloxanes, and carbosilane dendrimers. All the radical hydrothiolation reactions presented in the review give beta-addition products. At the same time, alpha-addition products are formed in minor amounts only. This applies both to reactions that involve compounds with allyl or vinyl substituents at the silicon atom and to reactions of 3-mercaptopropylsilane derivatives with olefins [28]. This gives an advantage to the hydrothiolation reaction compared to hydrosilylation, since the latter can produce a significant fraction of alpha-isomers. On the other hand, hydrosilylation may be preferable in reactions with vinylsilanes.

Yet another aspect of the hydrothiolation reaction that should be noted is that it is accompanied by a side reaction that gives disulfides, the fraction of which tends to increase in the presence of oxygen. The presence of oxygen and an oxide catalyst in the system is a necessary condition for the synthesis of disulfides [29]. A considerable number of studies on hydrothiolation are performed with silanes containing hydrolytically unstable alkoxy groups [28,30,31], which necessitates the use of anhydrous solvents. It should be remembered that the solubility of oxygen in nonpolar solvents is very high, so oxygen should be preliminarily removed from a solvent in operations with such silanes. It was noted separately in the work mentioned above [26] that no side processes occur in the reaction of various thiols with tetravinylsilane if the reaction is carried out in methanol, though it occurs in air. The adverse effect of oxygen on the rate of the process is also reported in another work [31]. This fact also makes the hydrosilylation reaction indispensable in some cases.

Below, this review provides specific examples of thiol-ene addition to organosilicon substrates. The overview of this issue will begin with individual low-molecular compounds and then continue to increasingly complex structures. In particular, examples of the hydrothiolation of POSSes and organosilicon macrocycles, then siloxane polymers, and finally examples of preparing materials of siloxane nature will be considered.

### 1.2. Modification of Individual Organosilicon Compounds by the Thiol-Ene Reaction 

The hydrothiolation of various organosilicon compounds can be used to obtain monomers for synthesizing functional polysiloxanes [32].

Commercially available mercaptopropylmethyldimethoxysilane (see Figure 6) is a very convenient precursor for synthesizing siloxanes that comprise various functional groups. It can be used both as a monomer and as a co-monomer in the preparation of functional polysiloxanes. It was found in studies on these polymers that, despite their complex composition and the presence of sulfur, they have satisfactory thermal stability (up to 260 °C) and feature an unusual Si-S coordination bond that has certain chromophoric properties. The existence of this interaction was also demonstrated in other works [33,34].

The synthesis of linear polysiloxane products is based on the use of dialkoxysilanes. Along with them, trialkoxysilanes are also widely used in polymeric products to create branched architectures. However, they are used much more often as surface modifiers of solid materials [18].

For example, Tucker-Schwartz et al. [28] performed the radical addition of thiols to allyl-trialkoxysilanes and 3-mercaptopropyl-trialkoxysilanes to various allyl-containing organic compounds. Beta-addition was observed in both cases (see Figure 7). They obtained a wide range of agents with trialkoxysilyl moieties that are rather promising surface modifiers for nanoparticles and materials based on SiO_2_ and transition metal oxides (this subject will be discussed below in more detail). Hydrolysis of alkoxy groups at silicon gives silanols that can react further to give Si-O-Si and Si-O-metal bonds. It is this feature of silanols that provides such a good binding to inorganic materials.

The inorganic nature of the Si-O-Si bond is often used to create organo-inorganic hybrid materials. Esquivel et al. [35] used radical hydrothiolation in the synthesis of a bridging bis-trialkoxy precursor (see Figure 8). Compounds of this kind are used for the subsequent synthesis of regular mesoporous organosilicates. They are usually made inert, but in this case the use of hydrothiolation made it possible to create functional organosilicates capable of further modification.

It is also possible to modify pre-manufactured organic materials with hybrid modifiers. Chinese scientists Fu et al. [30] synthesized such a modifier from mercaptopropyltrimethoxysilane and castor oil (see Figure 9). This modifier is one of the starting materials for the further preparation of hybrid coatings based on polyurethane with Si-O-Si cross-links. The thermal resistance, strength, and hydrophobicity of the resulting coatings can be improved by increasing the number of these cross-links. It should also be noted that in this case, addition to internal rather than terminal double bonds occurs, which would be very difficult in the case of hydrosilylation of these substrates.

To date, trialkylsilyl-containing organosilicon modifiers and precursors have long since ceased to be a subject of purely basic studies and appear quite often in various patents. For example, this pertains to modifiers based on di- and trialkoxysilanes with an isocyanate group bound through a thioalkyl bridge. Such compounds can be used as an adhesive layer on polarized films that are utilized in the manufacture of LCD displays of modern gadgets. They are also compatible with many elastomeric materials and can provide additional cross-linking and impart self-strengthening properties under high humidity and high temperature conditions [36].

The synthesis of organosilicon derivatives without alkoxy groups also opens up access to a wide range of functional compounds. Though the application area of these compounds is much less obvious, their synthesis is also interesting from the scientific point of view.

The hydrothiolation of tetravinylsilane with various thiols can be used to obtain individual tetrafunctional compounds [26] (see Figure 10).

By varying the nature of R, a wide range of functional compounds can be obtained in one step, in good yields and without complex isolation. Depending on the functional groups, they can serve as polydentate ligands, cores for dendrimers, stabilizers of complexes, etc.

Yet another synthesis of quite interesting compounds is given in the works by Trofimova et al. [31,37]. They aimed at creating organosilicon compounds that comprise benzoxazole and benzothiazole heterocyclic moieties. The synthesis was performed for research purposes only, and the authors do not comment on the potential applications of the compounds obtained. However, this example again demonstrates the high tolerance of the hydrothiolation method to various functional groups (see Figure 11).

It is worth to mention, the benzothiazole derivatives can also be prepared by hydrosilylation of C=N bond [38].

As one can see from the above, the combined presence of alkoxy groups and functional groups bound to a silane opens up wide opportunities for creating new materials and modifying existing ones. The number of alkoxy groups can vary from one to three, depending on the purpose. Examples of such structures and their applications are presented below.

If a compound contains two alkoxy groups, it is possible to synthesize a functionalized monomer for the subsequent preparation of linear functional siloxanes, as shown above. For example, Cao et al. [32] first hydrotiolated allyl-methyl-dialkoxy-silane with various mercaptans containing polar groups and then used the resulting compound in hydrolytic polycondensation (see Figure 6). This synthetic approach opens a way to the simple and convenient preparation of polysiloxane amphiphiles that are in demand in agriculture.

Currently, monomeric amphiphiles with a small but controlled siloxane part are also in demand. They are usually obtained by hydrosilylation of allyl-PEG with trisiloxanes that carry hydride at the central silicon atom [39]. Combining Piers–Rubinsztajn reactions with thiolation makes it possible first to create a nonpolar controlled siloxane structure and then attach a polar fragment by hydrothiolation [40] (see Figure 12).

A similar approach was implemented to obtain water-soluble monomeric siloxane surfactants. Reprinted/adapted with permission from Ref. [41].

Moreover, the combination of hydrothiolation with subsequent hydrosilylation looks quite promising, though it has not yet been studied in sufficient detail. It was shown convincingly [42] that upon hydrothiolation of silanes containing hydride at silicon, this Si-H bond is preserved and is theoretically suitable for subsequent hydrosilylation (see Figure 13). 

So far, the combination of these two reactions was used to synthesize telehelic bifunctional polymers based on polystyrene [43] and cross-linked polymeric materials [44].

Also, there are some examples of using this reactions in reverse order. Octakis(dimethylvinylsiloxy)octasilsesquioxane derivatives with different functional groups were synthesized by incomplete Hs with subsequent Ht [45] Moreover, an interesting work on the sequential modification of limonene was published [46]. The different reactivity of the terminal and internal double bonds in limonene was used to perform its consistent and selective hydrosilylation and hydrothiolation. A few pages above, we already mentioned the hydrothiolation of internal bonds in castor oil. Both of these cases clearly show yet another advantage of hydrothiolation over hydrosilylation, namely, a higher reactivity of the thiol group in comparison with the Si-H group. This work will be discussed below in more detail.

The subject of sequential application of hydrothiolation and hydrosilylation of silanes and individual siloxanes containing both double bonds and Si-H groups remains open to date.

Thus, the thiol-ene reaction is a convenient tool for modifying alkoxysilanes. The tolerance of the Ht (hydrothiolation) reaction makes it possible to introduce a wide variety of functional groups into alkoxysilanes. Further, depending on the goal, alkoxy groups can be subjected to hydrolysis and condensation (see Figure 14). The use of a bifunctional alkoxysilane makes it possible to obtain linear polymers and copolymers. Trialkoxysilanes can act as surface modifiers or precursors of hybrid organic-inorganic materials.

On the other hand, the Piers–Rubinsztajn reaction (see Figure 12) makes it possible to obtain alkoxysilanes with a well-defined structure. Compounds of this kind are promising as surfactants, for example.

Finally, a combination of Ht and Hs (hydrosilylation) reactions appears promising (see Figure 14). For example, first Hs and then Ht are performed in the case of limonene. Or other cases, Ht addition to a silane with preservation of a SiH group can be used.

### 1.3. Modification of Cyclic and Polycyclic Organosilsesquioxanes Using the Hydrothiolation Reaction for Creating New Supramolecular Systems and Materials Based Thereon 

To date, polyhedral organosilsesquioxanes (POSS) are of particular interest among organosilicon compounds. Over the past five years, 3293 works on this subject have been published. This is due to the fact that POSS are unique organo-inorganic matrices. They are also used as nanoscale building blocks and fillers in the preparation of hybrid polymeric materials. The keen interest in POSS is caused by their potentially wide practical application in various areas of materials science; they are also interesting from the theoretical point of view [47,48,49,50,51,52].

The functional POSS that are used most widely include pentacyclo-[9.5.1.1^3,9^.1^5,15^.1^7,13^]-octasiloxane, 1,3,5,7,9,11,13,15-octaethenyl (**1**) [53,54,55] and pentacyclo-[9.5.1.13,9.15,15.17,13]-octasiloxane-1,3,5,7,9,11,13,15-octapropanethiol (**2**) [56,57,58,59] (see Figure 15).

This is due to the simplicity of synthesizing, availability of the starting reagents, and synthetic potential of the functional groups. Numerous works on the modification of vinyl-containing POSS by hydrosilylation have been published to date [47]. Various POSS derivatives and materials based on them can be obtained in this way. One of the examples involves the preparation of heat-resistant composites derived from polyhedron **1** and a hydride-containing methylsilsesquioxane resin by hydrosilylation in the presence of Karstedt’s catalyst (see Figure 16).

Moreover, it should be mentioned that significant progress in the hydrosilylation of S, N-containing alkenes has been achieved [60]. However, the reported procedure requires a high Ir loading when compared to the typical Pt-hydrosilylation procedures. An efficient solution was found by Marciniec’s team [61] who used two reactions for the functionalization of polyhedron **1**, i.e., cross-metathesis and silylative coupling (Figure 17).

This approach made it possible to incorporate substituents containing heteroatoms (O, N and S) into POSS structures.

The use of the hydrothiolation reaction appears to be one of the solutions to this problem that allows both functional POSS **1** and **2** that are most accessible to be utilized (see Figure 15). The advantages of this approach include numerous commercially available starting compounds, mild synthesis conditions, the possibility of performing the reaction without a catalyst, etc. Over the past five years, over 80 papers on the use of the hydrothiolation reaction for modification of POSS and production of materials based on them have been published.

Due to good thermal characteristics and high functionality, POSS is often used for the production of composite conductive materials. The combination of these factors made it possible to make lithium-ion batteries [62] and obtain ionic liquids [63,64] based on POSS and hydrothiolation reactions. Good conductive and thermal properties of the target materials are noted in publications.

The creation of omniphobic [65] and superhydrophobic [66,67] coatings for various materials on their basis was found to be an interesting field of POSS application. The hydrothiolation reaction proved to be very successful for these purposes due to its advantages described above. An important feature of this approach is the possibility to obtain hard surface layers characterized by high hydrophobicity and thermal stability. It was shown [65] that a highly efficient protective coating can be obtained based on POSS **1** using the hydrothiolation reaction and perfluorinated organics (see Figure 18).

Moreover, materials obtained by this methodology are used as sponges for water removal from oil [66,68,69].

A new category of amphiphilic molecules called giant surfactants appeared in recent years [70]. They attract much attention due to their unique self-assembly behavior, both in the bulk and in solutions. The variability of chemical modification and precise control of the molecular topology endow giant surfactants with a complex architecture, thus leading to a more complex self-assembly behavior and tunable functional properties.

Two main approaches are available to obtain POSS containing surfactants. The first one involves the synthesis of octa-substituted addition products (star polymers [71,72], polyfunctional surfactants [73], and crosslinked fibers [74]). The second approach is based on synthesizing a monofunctional derivative by hydrothiolation followed by modification of the remaining seven functional groups using the same reaction [75,76,77,78]. A striking example of such a modification is presented in [78] (see Figure 19).

The very fact that a monofunctional product can be obtained in the presence of eight reactive groups in a structure is surprising. This fact made it possible to obtain efficient giant surfactants (see Figure 20).

Much attention has been recently paid to materials that can selectively sorb certain compounds from water. This is relevant for water purification from compounds that can harm the environment. The use of the hydrothiolation reaction for POSS modification made it possible to obtain efficient composite sorbents for the adsorption of silver ions [79] and organic dyes (methylene blue) [80] from water.

The development of new medical drugs is an important task in modern human life. POSS can also be used in this field. Compounds of this class are hydrophobic. Therefore, they have to be modified with hydrophilic groups for application in aqueous media. Hydrothiolation is an ideal reaction for this purpose. For example, PEGs were grafted onto POSS to make them soluble, and the BODIPY derivative was used as a photosensitizer for photodynamic/photothermal therapy [81]. In another study [82], hybrid oligo(ethylene glycol)-polyhedral silsesquioxane materials were obtained. These composites demonstrate excellent bioactivity with formation of hydroxyapatite whose morphology depends on the molecular weight of the ethylene oxide spacer. Thus, these systems can be applied in practice for bone regeneration, which may be of great importance for bone tissue engineering in the future.

Apart from POSS, the synthesis of cyclic silsesquioxane structures has been actively developing recently. From the synthetic and structural points of view, they appear even more interesting objects for the development of new materials. Currently, compounds with various framework structures and functionality can be obtained on their basis. A great contribution to the development of this approach was made by the Unno team [83,84,85,86,87,88,89,90,91]. They synthesized compounds with diverse structures and functionality, in particular, Double-Decker siloxanes [85,87,91]. It appears that the following structures are most interesting in terms of hydrothiolation usage (see Figure 21).

Shchegolikhina’s team is actively developing the synthesis of stereoregular organosiloxanes, a unique class of organosilicon compounds [93,94,95]. These compounds are synthesized from polyhedral metallosiloxanes and cannot be obtained by the classical organosilicon chemistry reactions. The possibility of modifying all-cis-tetravinyl-tetrakis(trimethylsiloxy)cyclotetrasiloxane and cis-tetravinyl-tetrakis(dimethylvinylsiloxy)cyclotetrasiloxane with various polar fragments by the hydrothiolation reaction was shown [96]. In particular, an amphiphilic “Janus” cycle was obtained (see Figure 22); compounds of this class are very much in demand in supramolecular chemistry because of their ability to undergo self-organization under various conditions.

Organo-inorganic hybrid materials have both the mechanical properties of organic polymers and the high thermal resistance and functionality of inorganic materials. The preparation of these materials often comes down to incorporation of special fillers into polymer matrices. An example of such a filler was synthesized in [93] by hydrothiolation of pentacyclo[9.5.1.13,9.15,15,17,13]octasiloxane with 1,3,5,7,9,11,13,15-octaethenyl-9-mercaptocarborane. As a result, a polycyclic silsesquioxane with eight carbonyl substituents in the structure was obtained (see Figure 23).

Based on all of the above, it can be concluded that the hydrothiolation reaction can be used even more actively in the preparation of new derivatives of cyclic and polycyclic organosiloxanes and materials based on them. This is confirmed by the increasing number of publications on this subject in recent years.

### 1.4. Modification of Siloxane Polymers by Thiol-Ene Chemistry Methods

Let us now consider the examples of using Ht reactions for organosiloxane polymers. The modification of siloxane polymers by thiol-ene chemistry methods is a popular technique in this field. As a rule, two main approaches are used. The first one involves the use of a vinyl-containing polysiloxane followed by modification with an organothiol (see Figure 24I). The second approach involves the use of a thiol-containing siloxane (see Figure 24II):

The starting polymers PSVin are prepared by ring-opening polymerization (ROP) of siloxane rings in octamethylcyclotetrasiloxane (D4Me) and tetramethyltetravinylcyclotetrasiloxane (D4MeVin) in the presence of a chain terminating agent (hexaorganodisiloxane). The number of links *n* is determined by the loading of the corresponding ring [97]. The molecular masses of the resulting polymers range from low-molecular-weight oligomeric products (~800 kDa) to 130 kDa. The terminal groups are usually either–SiMe_3_ or –SiMe_2_Vin, depending on the choice of the chain terminating agent at the ROP stage. In addition to PSVin with distributed vinyl links, PDMS with –SiMe_2_Vin terminal groups are also used [98]. The use of PSVin with nonlinear architectures in the Ht reaction is covered in the literature much more scarcely. Examples where a ladder polyvinylsecquioxane is used are reported [99]. An example of preparing a carboranesiloxane dendrimer by alternating thio-Michael and radical Ht reactions already mentioned above was demonstrated [16].

A few options are possible in the case of PSsh. As a rule, gamma-propylmethyldimethoxysilane is used as the starting reagent. In this case, both its co-hydrolysis with a dimethyldialkoxysilane and catalytic rearrangement with D4Me are performed. Co-hydrolysis with mercaptopropyltrimethoxysilane results in branched PSsh [100]. It is worthy of note that the molecular mass of the resulting polymers is rather small, i.e., 0.8 to 9 kDa. Therefore, the use of such siloxanes cannot provide higher physical and mechanical properties compared those of vinyl or hydride analogues. From this point of view, it is preferable to use PSVin. At the same time, if PSsh is used, alkynes can be used for double addition [101].

In the case of PSVin and PSsh, usually R = Me. This is primarily due to the availability of the starting monomers and their reactivity in the synthesis of PSVin and PSsh polymers. Cole and Bowman [102] compared the reaction kinetics of Ht on phenyl-containing PSsh copolymers.

As a rule, the Ht of the polymers in question is performed photochemically or by thermal initiation. DMPA is most commonly used as the photoinitiator, while sometimes benzophenone is used. The reaction is performed at room temperature and takes a few minutes.

Thermal initiation is most commonly performed with AIBN. The reaction is carried out at 65–85 °C.

As mentioned above, the main advantage of the Ht approach over Hs is that the Ht reaction is tolerant to functional groups of diverse nature and reaction conditions. At the same time, the method is versatile, i.e., regardless of the group incorporated into a siloxane, the reaction conditions may remain the same. For example, it was shown [103] that introducing a large number of polar groups can be incorporated into PSVin under the same conditions (see Figure 25).

Yet another advantage is the greater selectivity of Ht in addition to the double bond. The Hs of polysiloxanes with functional compounds often gives a mixture of products with various structures that are rather difficult to separate [104]. The properties of such mixtures are hard to control and unpredictable. In the case of Ht, the addition predominantly occurs against Markovnikov’s rule. This is important, for example, in the synthesis of self-organizing polymers such as amphiphilic [40,105] or mesomorphic ones [106,107,108].

Apart from self-organizing systems, other smart materials, in particular, self-healing ones, are also obtained by the Ht reaction [34,109]. This effect is reached by introduction of groups that can form donor-acceptor bonds, for example, those containing boron and nitrogen. Incorporation of groups that change conformation upon UV irradiation makes it possible to obtain photosensitive materials [110].

The presence of sulfur in the target polymers causes both drawbacks and advantages. The obvious drawback is that the thermal-oxidative stability of sulfur-containing polymers is lower than that of their sulfur-free analogues. On the other hand, the presence of sulfur allows the adhesion to metals to be increased. For example, the interaction of gold with polysiloxanes modified with thiols of various nature was studied [111]. In particular, the modification of polyvinylmethylsiloxane with mercaptoacetamide made it possible to obtain water-soluble products. Yet another work [107] demonstrated the unconventional chromophore S→Si coordination bonds that were originally found in water-soluble comb polysiloxanes with different ratios of polyether and mercaptopropyl groups as side chains.

Moreover, the so-called anchor groups can be incorporated by the Ht reaction. These groups make it possible to further coat various surfaces with a modified polysiloxane. Depending on the nature of the anchor group, both chemical and donor-acceptor interaction with the surface is possible. In [112], gamma-propyltrimethoxysilane grafted onto PSVin served as the anchor group. Subsequently, chemical modification of cellulose surface occurs via alkoxy groups. After that, the surface becomes superhydrophobic (the contact angle is 154°). An example of chelate interaction of an anchor group with the surface of ZnO particles is provided in [113].

Apart from siloxanes, other organoelement polymers with unsaturated bonds can also be modified. In [114], mercaptopropyltrimethoxysilane was grafted onto linear polyphosphasene. Further, hybrid nanoparticles that are promising flame retardants and biomedical materials were obtained by the sol-gel method (see Figure 26).

In another example [115], an acrylonitrile-butadiene rubber was modified with a fluorosilicon rubber (MNBR/FSR). The properties of blends of MNBR/FSR and unmodified acrylonitrile-butadiene rubber (NBR) with a fluorosilicon rubber (FSR) were compared. The physical and mechanical properties of MNBR/FSR were found to be strongly superior to those of NBR/FSR. For example, the tensile strength of MNBR/FSR is 14.34 MPa versus 2.92 MPa in NBR/FSR.

It is worth mentioning that in the last example, NBR (1,4-addition) was used for the modification. Double bonds in such polymer are internal and, as a consequence, poorly react in the hydrosilylation reaction, what was mentioned above. Accordingly, when using polybutadiene (1,4-addition) the hydrothiolation reaction is preferred. On the other hand, during the reaction of polybutadiene containing a large amount of vinyl groups (1,2-addition), a side reaction of intramolecular cycloformation proceeds to a large extent [116,117]. In this case, the hydrosilylation reaction is preferred, what was demonstrated in the works [118,119].

Thus, the approaches considered above make it possible to obtain polysiloxanes with diverse functional groups that determine the final product properties. In the future, these polymers can be used both independently and in composite materials. The number of functional groups that can be grafted varies widely. This allows for further crosslinking reactions to be performed via the residual thiol or vinyl groups. The preparation of materials by the Ht reaction is widely covered in the current literature and will be discussed below in this review.

### 1.5. Modification of Polysiloxanes with Some Large-Tonnage Biomass and Waste Compounds

The modification of natural compounds is discussed in a separate chapter because the polymer science has recently been considering the prospects of utilizing renewable natural raw materials as an alternative to large-tonnage monomers that are mainly obtained from oil and gas processing products. The first reason is that attempts are made to reduce the amount of pollution arising in the production of the base monomers, their polymerization and further processing, and disposal of polymer wastes. Second, renewable natural raw materials are a cheap and accessible source of starting reagents.

It should also be noted that biomass compounds that are useful for polymer chemistry often present a problem for recycling. A few most representative examples of large-tonnage sources of starting compounds for polymer synthesis can be provided. Lignin, which is a complex mixture of polyphenols and benzoic acids that can be isolated individually, is one of them. Nevertheless, lignin is still one of the large tonnage wastes of the pulp and paper industry. Soybean oil, which mainly consists of fatty acid esters, is another example. Since soybeans are grown on a large scale, their processing produces tons of waste, so it is also interesting in terms of their use as a source of useful chemical reagents. For example, 100 g of soybean oil contains about 16 g of monounsaturated esters, 23 g of monounsaturated esters, and 58 g of polyunsaturated esters [120]. Terpenes are the third most important example. Pinene and limonene are the most abundant of them. Various forms of pinene are found in the oleoresins and turpentines obtained from conifers, while limonene is found in citrus peels. Hence, these terpenes can be extracted from the wood and food industry wastes by well-proven methods [121,122]. Currently, 50–75 million kg of limonene is produced per year. It is already used as an organic solvent, a non-toxic alternative to hexane and cyclohexane [123]. It should also be emphasized that nearly all monoterpenes are built as a combination of isoprenyl moieties, in particular, limonene consists of two such moieties. Since isoprene rubber can be considered as an isoprene derivative, pyrolysis causes its depolymerization to isoprene, a fraction of which further dimerizes to yield limonene. Moreover, it was shown that butadiene rubber can also undergo pyrolysis under certain conditions to give fractions containing limonene [124]. Though works in this field are at an initial level only, they create an important precedent in the use of terpenes as important raw base materials that can be processed further.

All of the above three classes of natural compounds have been the subject of keen interest to researchers, including those from organic and polymeric chemistry. The use of the hydrosilylation and hydrothiolation approaches was not an exception. They became relevant due to the availability of multiple bonds that undergo these two reactions in all the above compounds, as well as the presence of other functional groups capable of further chemical transformations.

The joint use of Hs and Ht reactions requires that a few conditions are met. For example, Hs reactions are usually performed in the presence of platinum catalysts whose action is inhibited by some functional groups, such as -NH_2_, -SH, and -COOH, in the substrates. Ht reactions are more tolerant to functional groups, however, in the radical variant of this reaction, groups that act as radical traps can become a hindrance. Therefore, approaches with protection of functional groups are usually employed. In addition, the Hs reaction is performed first, followed by the Ht reaction. This scheme can be illustrated by the following example (see Figure 27).

At the first stage, methyl undecenoate is hydrosilylated with tetramethyl disiloxane, while at the second stage, transesterification with undecenoic alcohol is carried out. At the final stage, the resulting diene is polymerized with the corresponding dithiol by an Ht reaction [125]. In [126] a series of siloxane-containing copolymers based on undecenoic acid diamide were obtained using either Hs or Ht reactions (see Figure 28).

The thermal and thermo-oxidative stability of the sulfur-containing analogues was shown to be lower than that of their analogues.

A simpler approach with the use of Ht and Hs reactions was demonstrated [46].

The idea of this approach relies on the diverse reactivity of limonene. Thus, it was shown that the isoprenyl double bond enters into the Hs reaction on Karstedt’s catalyst, while the cyclohexene double bond remains non-reactive. However, the latter enters the Ht reaction (see Figure 29a, compound 3). Firstly, this approach was demonstrated in this example and then in the synthesis of copolymers (see Figure 29b). It is possible to perform the first Hs reaction to obtain AA type monomers, and then perform their polyaddition with dithiols by the second Ht at the final stage [127]. Another approach was also shown: based on limonene, a dicarboxylic derivative is first obtained by the Ht reaction and then used in a polycondensation reaction with the corresponding diamine, which leads to the formation of polyamides [128].

An interesting approach was suggested for carvone, a limonene analog. Carvone is characterized by the presence of a keto group in the cyclohexene ring, while the keto group forms a conjugated system with the double bond. It should be expected that this compound would enter nucleophilic Michael reactions, as shown by Drozdov et al. [129] (see Figure 30).

Moreover, it was found that carvone forms two different addition products with dithiol depending on the reaction conditions. In fact, if the reaction is carried out in the presence of a base, a Michael addition product is obtained. However, if the reaction is performed under UV light irradiation in the presence of a sensitizer, addition to the isoprenyl double bond occurs.

Yet another approach involves the simultaneous functionalization and cross-linking of PDMS with distributed methylhydridosilyl units by the Hs reaction that occurs with tris(pentafluorophenyl)boron as the catalyst (see Figure 31) [130].

It was shown that in the presence of tris(pentafluorophenyl)boron, hydridosilanes react with carbonyl compounds at much higher rates than with double bonds, and this fact was used here. At the final stage, Ht of thiolcoumarin dyes to the functional double bonds of the cross-linked PDMS matrix is performed.

A specific application of a combination of Hs and Ht reactions was reported [124]. Synaptotagmin I C2Am (a marker of apoptosis in eukaryotes) was first converted to the corresponding propargyloxytriethylene glycol derivative OP-C2Am (see Figure 32).

Subsequently, the Hs of an organosilicon derivative of hippuric acid was performed under mild conditions (37 °C) in phosphate buffer using small amounts of a Ru(II) catalyst. It should be noted that under these conditions, hydrosilylation occurs even if the substrate comprises a sulfur atom.

Thus, some conclusions on the use of the hydrosilylation and hydrothiolation methods to obtain functional derivatives and copolymers based on natural renewable raw materials can be made. The thiol-ene addition to multiple bonds in many natural substrates was widely used, though the hydrosilylation was only used by some scientific teams. Nevertheless, it has been shown that polymers obtained by the hydrosilylation reaction are more thermally stable and less prone to thermal oxidative degradation due to the absence of sulfur atoms in the polymer chain. The combination of hydrosilylation and hydrothiolation shows good prospects for various applications in the field of polymer chemistry for synthesizing both copolymers and cross-linked polymers.

### 1.6. Cross-Linked Organosilicon Polymers

The search for fast and soft vulcanizing methods is one of today’s main challenges in the chemistry of organosilicon polymers. The main drawbacks of traditional methods include the high cost of the platinum catalyst in the platinum-catalyzed hydrosilylation and the high process temperature in the case of the peroxide-initiated radical reaction. The production of silicone elastomers by the mild hydrothiolation reaction can be competitive to the traditional methods.

Cross-linked organosilicon polymers are widely used as materials for various purposes, namely as composites, coatings, biomedical materials, etc. These polymers are interesting as materials for biomedical applications due to their unique properties such as physiological inertness, low toxicity, and elasticity close to that of soft biological tissues.

The photoinitiated hydrothiolation reaction of a thio-functionalized polydimethylsiloxane with a vinyl-containing polydimethylsiloxane telehelix at room temperature is efficient for synthesizing crosslinked silicone elastomers (see Figure 33) [132]. Crosslinking occurs very quickly even in the presence of oxygen. These elastomers can be used for biomedical purposes and for cell cultures because they have good biocompatibility. A detailed study on the crosslinking of functional polydimethylsiloxanes is described elsewhere [133]. Biodegradable cross-linked polysiloxanes are promising objects for bioengineering applications [134].

A cross-linked porous polydimethylsiloxane promising for application as an acoustic material was obtained by thiol-ene addition in emulsion (see Figure 34) [135]. It was shown to have a very low sound velocity (∼40 m/s).

Polymer networks cross-linked by hydrothiolation and containing reversible covalent bonds due to dimerization of anthracene moieties present a good example of recyclable silicone materials (see Figure 35) [136].

Yet another example of recyclable materials based on thio- and vinyl-functionalized organosilicon compounds is given by self-healing polysiloxane elastomers with improved mechanical strength and good transparency containing esters of boronic [34], diboronic acids [137], or a combination of carboxy and amino groups (see Figure 36) [138] as the moieties responsible for the polymer network rearrangement. Moreover, combining hydrothiolation and Diels-Alder reactions applied to furan and maleimide organosilicon derivatives make it possible to obtain recyclable self-healing polysiloxane networks [109].

Fast photoinitiated hydrothiolation, in contrast to hydrosilylation, allows functional siloxanes to be used as precursors to obtain objects with complex structure based on silicone elastomers in 3D printing (see Figure 37) [139], imprint lithography [140], and photolithography [127].

Photo-curable polymers are widely used in the production of materials for electronic devices. For example, cross-linked sulfur-containing silicones with good insulating capability and high dielectric constant are promising for application as electrical insulators [141], while new hybrid materials obtained by UV-initiated hydrothiolation from thiol and vinyl phenylsiloxane precursors have high transparency in the visible wavelength range, high refractive index, good thermal stability, and extremely good insulating ability (see Figure 38) [142], which makes them promising for application as dielectrics in organic transistors [143].

The use of hydrothiolation makes it possible to produce silicone materials for optics and photonics, such as transparent glasses with a large refractive index, under mild noncatalytic conditions. In spite of the fact that many cross-linked organosilicon polymers with large refractive indices were obtained by the hydrosilylation reaction, the highest refractive indices can be achieved only in elastomers obtained by hydrothiolation due to the high atomic refraction of sulfur. Examples of such hybrid organic-inorganic polymers include thermoplastic UV-curable polyphenylsilsesquioxanes resistant to mechanical contact damage that have a tunable refractive index in the range of 1.467–1.546 (see Figure 39) [144]. Even better refractive indices (up to 1.703) were achieved in organo-inorganic hybrid polymers based on organosilicon, organogermanium, and organotin precursors containing double bonds, and in alkyl- and arylthiols [145].

Other examples of silicone materials for optical applications are luminescent organosilicon polysiloxanes based on grafted lanthanide complexes with N-acetyl-L-cysteine [146] and carboxylic acids [147]. It has been shown that this method can be used to obtain luminescent elastomers for application in light emitting diodes [148,149] and photochromic films [150]. The use of UV-curable hybrid polysiloxane materials is a promising approach for the production of flexible optical waveguides [151] and antireflection coatings [152].

The hydrothiolation reaction for surface modification finds use in the production of new composite materials for catalysis, chromatography [153], including the capillary columns for enantioselective nano-HPLC [154,155], for improving the strength characteristics of rubbers [156], and for biomedical purposes in the immobilization of enzymes such as chemotrypsin [157] and acetyl cholinesterase [158].

The prospects of the use of hydrothiolation in the production of nanocomposites based on organosilicon precursors are also worthy of note [159].

Solid polymer electrolytes based on polysiloxanes obtained by thiol-ene addition are promising thermostable materials for creating solid-state lithium-ion batteries on their basis (see Figure 40) [160].

Elastomeric membranes based on copolymers of polydimethylsiloxane and polyethylene glycol obtained by thiol-ene addition are promising objects for application in industrial gas separation modules (see Figure 41) [161,162]. These membranes possess good CO_2_/N_2_ selectivity, thermal stability and high mechanical strength characteristics.

The use of radical-initiated thiol-ene addition to give superhydrophobic polysiloxane aerogels directly in supercritical fluid CO_2_ was successfully demonstrated (see Figure 42) [163]. The main advantage of this method is that the drying stage can be eliminated from the overall process, which saves time and resources. Functional siloxanes [164,165] or phosphazenes [166] are used as the starting reagents.

Apart from other applications, the thiol-ene photoaddition method was successfully used to produce polysiloxane monodomain liquid-crystal elastomers [167] and other polysiloxane liquid-crystal materials [168].

## 2. Conclusions

In this review, we attempted to analyze the advances in the use of thiol-ene addition reactions in the chemistry of organosilicon compounds over the past 20 years as comprehensively as possible. A critical analysis was performed by comparing the hydrothiolation and hydrosilylation reactions as applied to the chemistry of silicones. It is worthy to note that in some cases, the use of hydrothiolation is considerably superior to hydrosilylation. This is particularly evident in cases where functional groups with a mobile hydrogen atom (amines, carboxy groups, alcohols, etc.) need to be incorporated in organosilicon substrates. We believe that the use of this reaction will continue to gain popularity with synthetic chemists. It is also important to note that this process can be used in combination with other reactions. All of these factors taken together allow us to state that in the near future, the development of all the approaches described in this review will lead to the development of new materials with a set of valuable physicochemical properties.

## Figures and Tables

**Figure 1 polymers-14-03079-f001:**
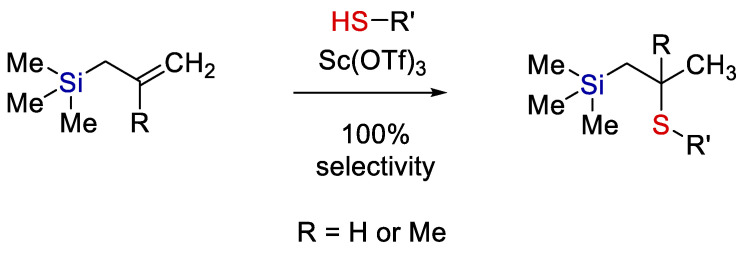
A scheme of ionic thiol-ene addition. Adopted from Ref. [15].

**Figure 2 polymers-14-03079-f002:**
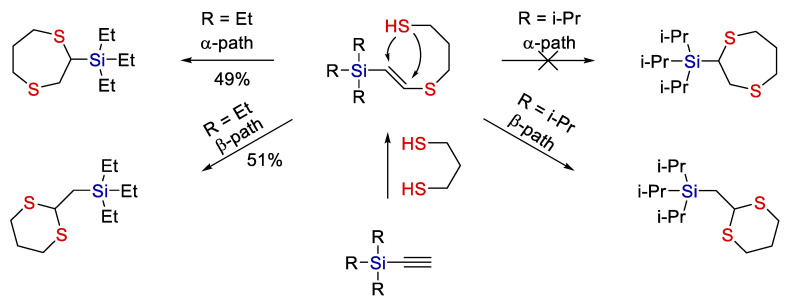
A scheme of ionic thiol-ene addition to ethynylsilane. Adopted from Ref. [15].

**Figure 3 polymers-14-03079-f003:**
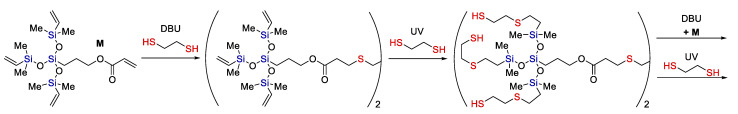
A scheme of the synthesis of carbosilane dendrimers by sequential Michael thiol-ene addition and radical thiol-ene addition reactions. Adopted from Ref. [16].

**Figure 4 polymers-14-03079-f004:**
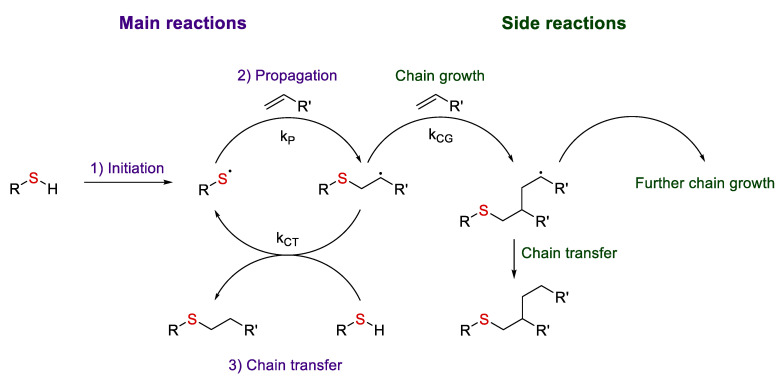
The mechanism of radical thiol-ene addition. Adopted from Ref. [17].

**Figure 5 polymers-14-03079-f005:**
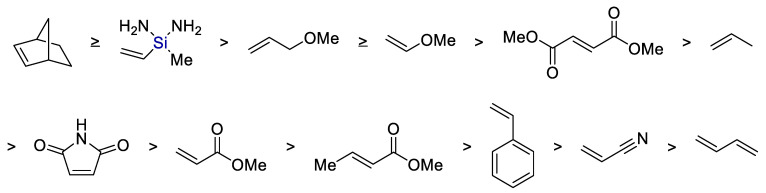
The predicted order of alkene reactivity with methyl mercaptan: norbornene ≥ vinyl silane > allyl ether ≥ vinyl ether > fumarate > propene > maleimide > methacrylate > crotonate > styrene > acrylonitrile > butadiene. Adopted from Ref. [20].

**Figure 6 polymers-14-03079-f006:**
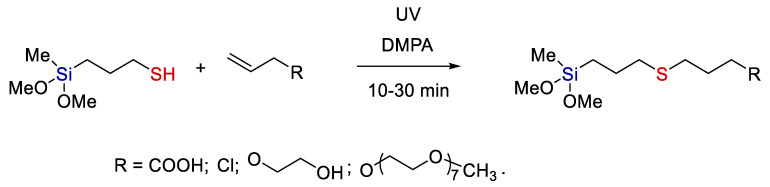
A scheme of the modification of mercaptopropylmethyldimethoxysilane by radical thiol-ene addition. Adopted from Ref. [32].

**Figure 7 polymers-14-03079-f007:**
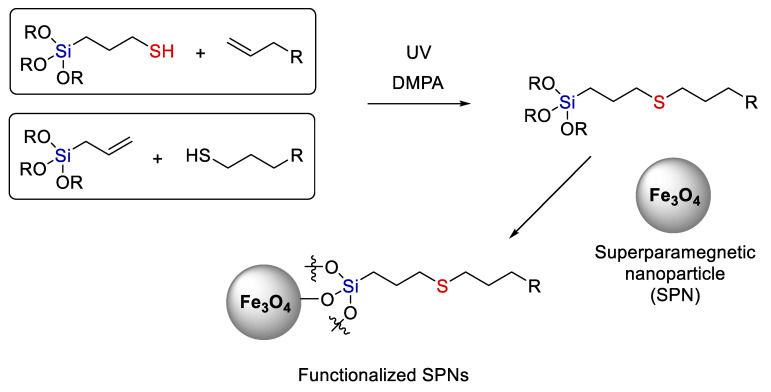
A scheme of modifying trialkoxysilanes by the thiol-ene reaction. Adopted from Ref. [28].

**Figure 8 polymers-14-03079-f008:**
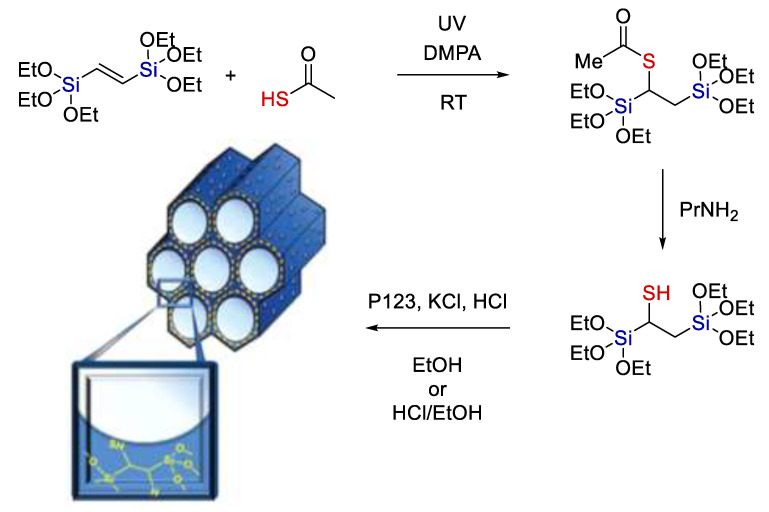
The synthesis of 1-thiol-1,2-bis(triethoxysilyl)ethane (2) and the corresponding thiol–PMO (SH–PMO). Adopted from Ref. [35].

**Figure 9 polymers-14-03079-f009:**
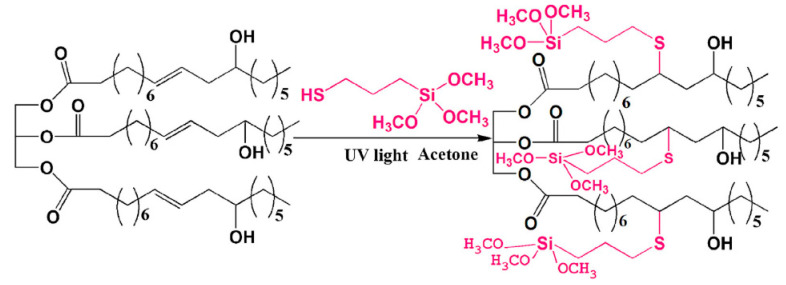
The modification of castor oil with mercaptopropyltrimethoxysilane. Reprinted/adapted with permission from Ref. [30]. Copyright (1997), with permission from Elsevier.

**Figure 10 polymers-14-03079-f010:**
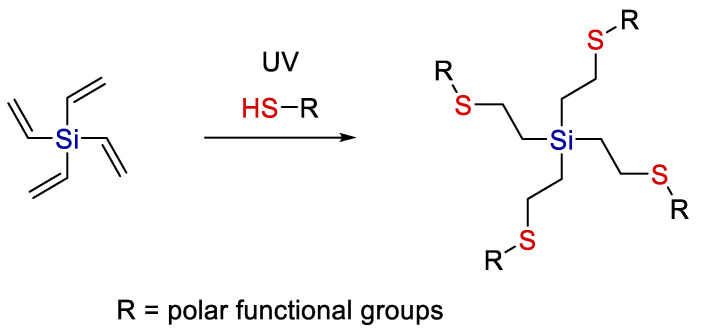
A scheme of modification of a tetrafunctional silane by thiol-ene addition. Adopted from Ref. [26].

**Figure 11 polymers-14-03079-f011:**
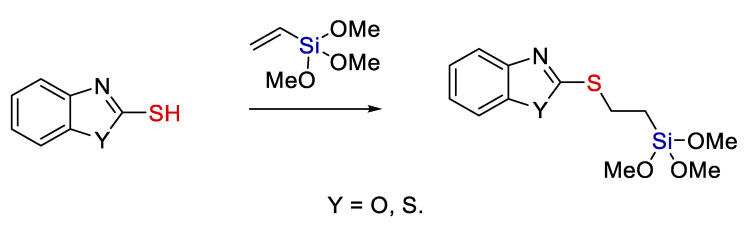
The synthesis of organosilicon derivatives of 2-mercaptobenzoxazole and 2-mercaptobenzothiazole. Adopted from Ref. [31].

**Figure 12 polymers-14-03079-f012:**
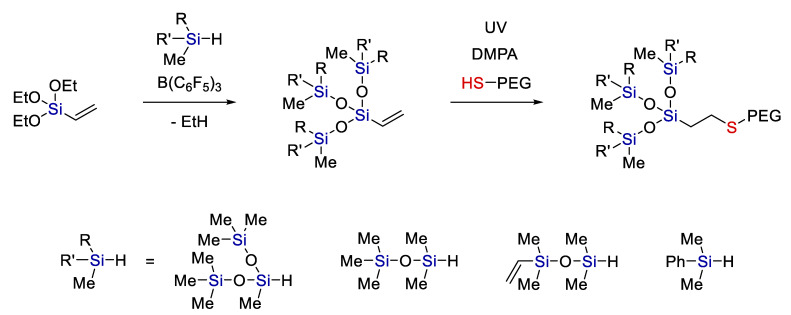
A scheme of consecutive Piers–Rubinsztajn and thiol-ene addition reactions. Adopted from Ref. [40].

**Figure 13 polymers-14-03079-f013:**
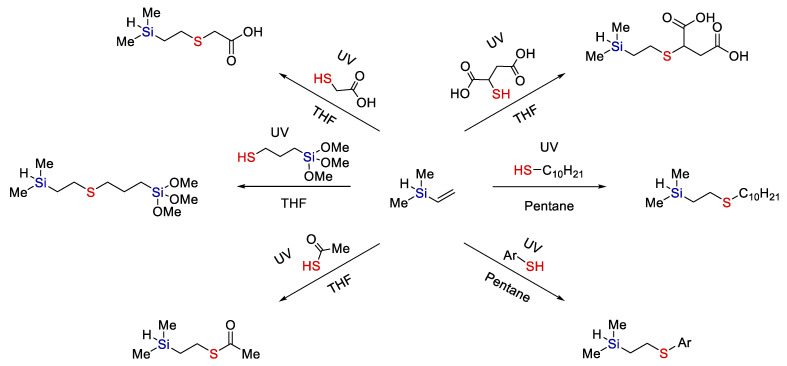
The synthesis of various functional silanes by thiol-ene addition with preservation of the Si-H bond. Adopted from Ref. [42].

**Figure 14 polymers-14-03079-f014:**
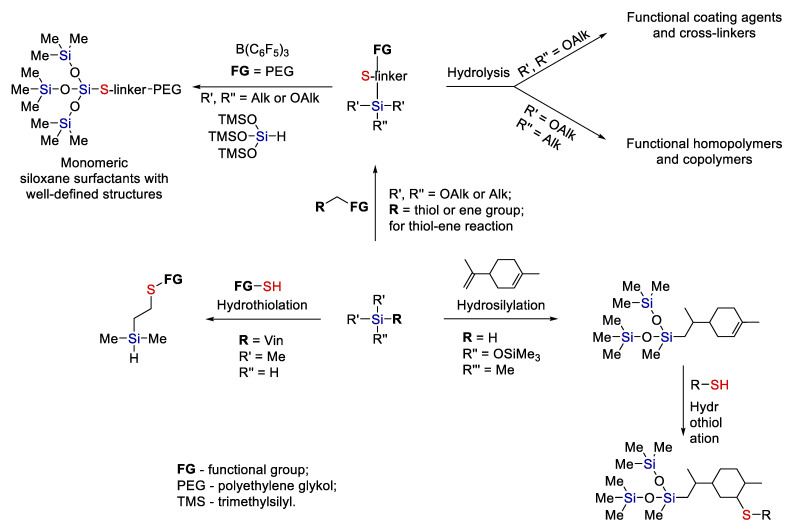
A general scheme for the modification of organosilicon compounds by thiol-ene addition combined with other synthetic approaches. Adopted from Refs. [32,38,40,43].

**Figure 15 polymers-14-03079-f015:**
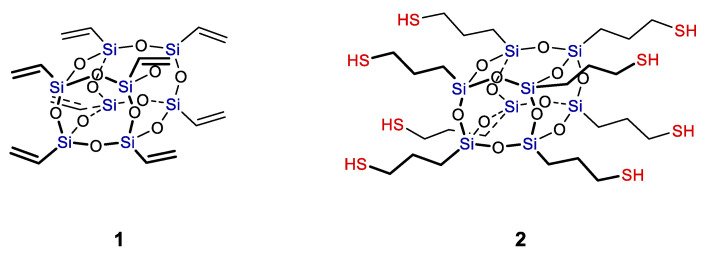
Pentacyclo-[9.5.1.1^3,9^.1^5,15^.1^7,13^]-octasiloxane, 1,3,5,7,9,11,13,15-octaethenyl (**1**) and Pentacyclo[9.5.1.13,9.15,15.17,13]octasiloxane-1,3,5,7,9,11,13,15-octapropanethiol (**2**).

**Figure 16 polymers-14-03079-f016:**
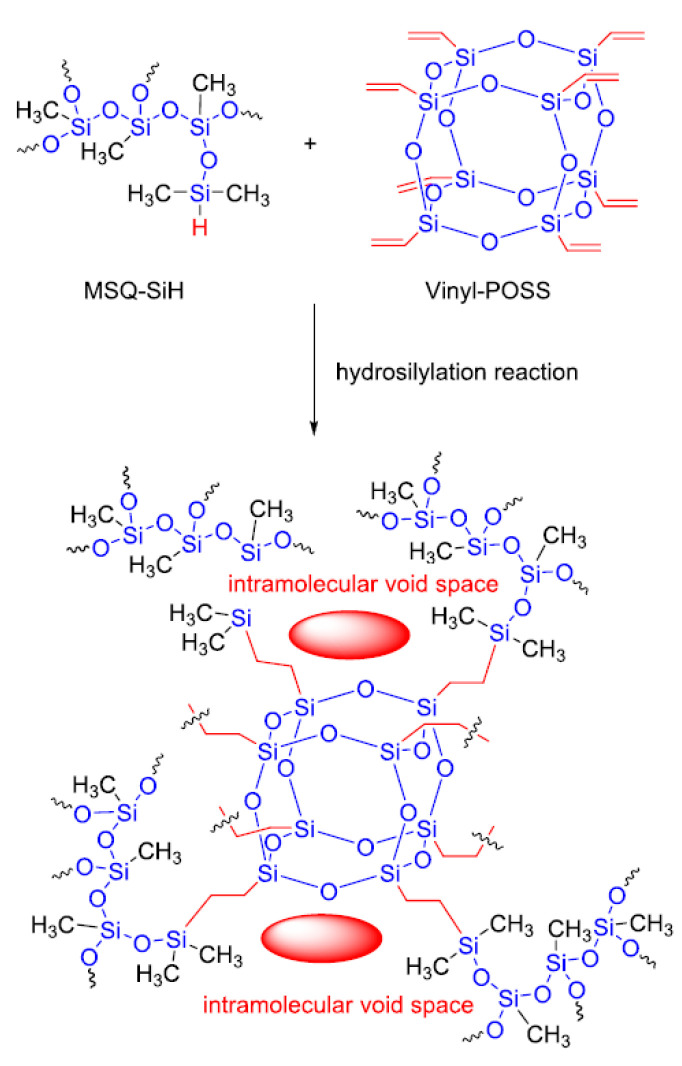
The hydrosilylation of hydrodimethyl-silylated oligomethylsilsesquioxane (MSQ-SiH) and octavinyl polyhedral oligomeric silsesquioxane (Vinyl-POSS). Reproduced with permission from Ref. [55]. Copyright 2021 American Chemical Society.

**Figure 17 polymers-14-03079-f017:**
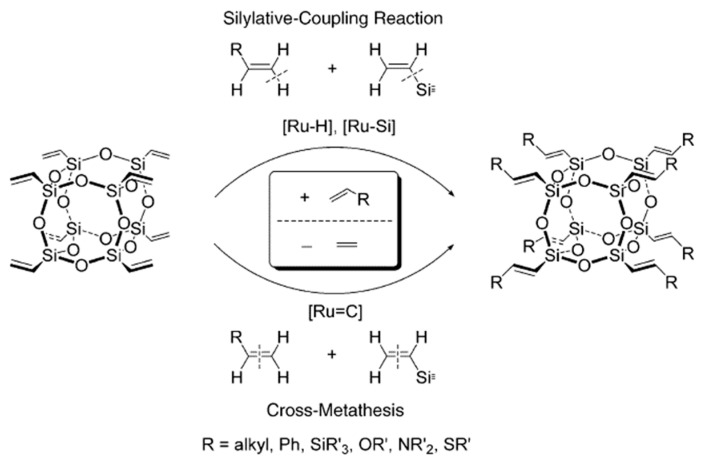
The combined reaction schemes for silylative coupling and cross metathesis. Reproduced with permission from Ref. [61]. Copyright 2004 John Wiley and Sons.

**Figure 18 polymers-14-03079-f018:**
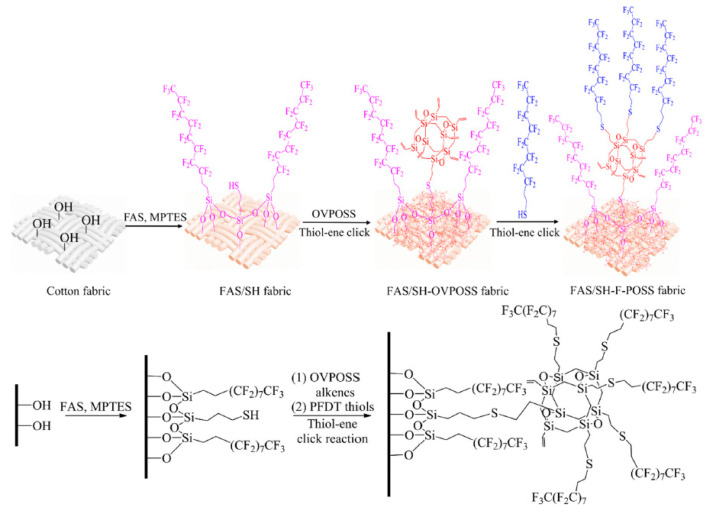
A schematic illustration of the procedure for preparing the FAS/SH-F-POSS coating on cotton fabric via silane coupling reaction and thiol-ene click reaction. Reproduced with permission from Ref. [65]. Copyright (2021), with permission from Elsevier.

**Figure 19 polymers-14-03079-f019:**
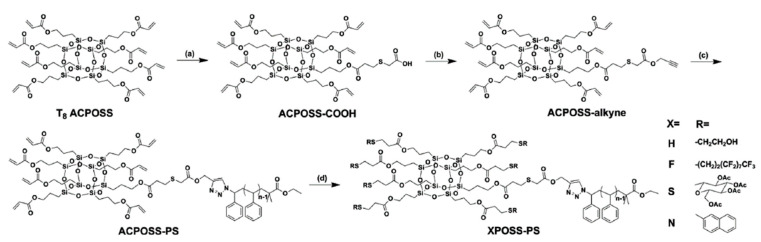
A synthetic pathway to giant surfactants using sequential “click” chemistry: (**a**) 2-mercaptoacetic acid, triethylamine, THF, 25 °C, 29%; (**b**) propargyl alcohol, DPTS, DIPC, dry DMF, 0 °C, 81%; (**c**) PSn-N3, CuBr, PMDETA, toluene, 25 °C, 83%–91%; (**d**) conditions I (thiol-Michael reaction): R-SH, hexylamine, THF, 25 °C, 0.5–2 h, 81%–93%; conditions II (thiol–ene reaction): R-SH, DMPA, THF, 25 °C, 0.5–2 h, 84–92%. Republished with permission of Royal Society of Chemistry, © 2014, from Ref. [78].

**Figure 20 polymers-14-03079-f020:**
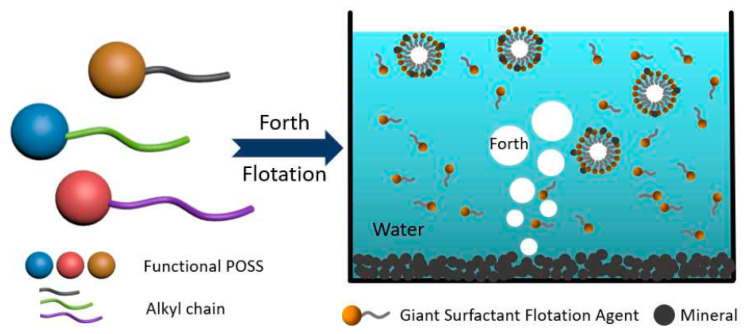
Forth Flotation Giant Surfactants. Reproduced with permission from Ref. [75]. Copyright (2019), with permission from Elsevier.

**Figure 21 polymers-14-03079-f021:**
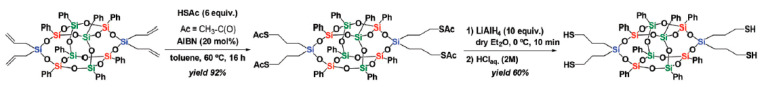
The synthesis of tetra-(3-mercaptopropyl)-DDSQ 3. Republished with permission of Royal Society of Chemistry, © 2021, from Ref. [92].

**Figure 22 polymers-14-03079-f022:**
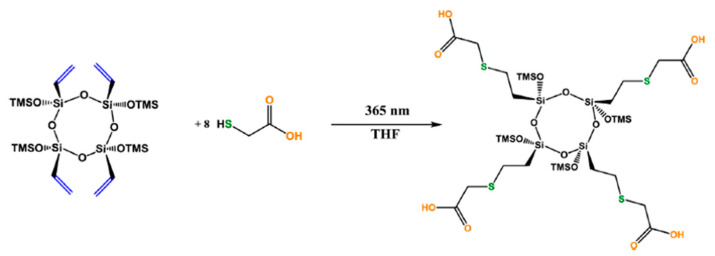
The reaction of all-cis-tetravinyl-tetrakis(trimethylsiloxy)cyclotetrasiloxane with thioglycolic acid. Reproduced with permission from Ref. [96]. Copyright (1997), with permission from Elsevier.

**Figure 23 polymers-14-03079-f023:**
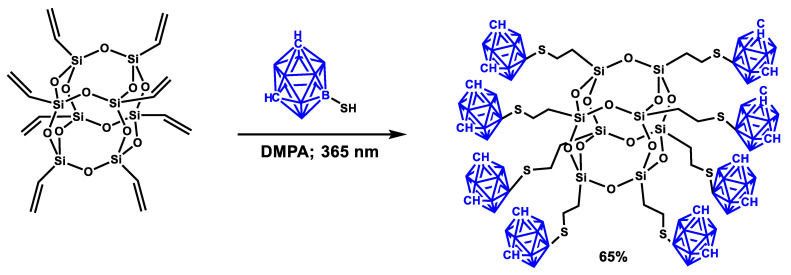
The scheme of synthesizing a carborane-containing POSS by thiol-ene addition. Adopted from Ref. [93].

**Figure 24 polymers-14-03079-f024:**
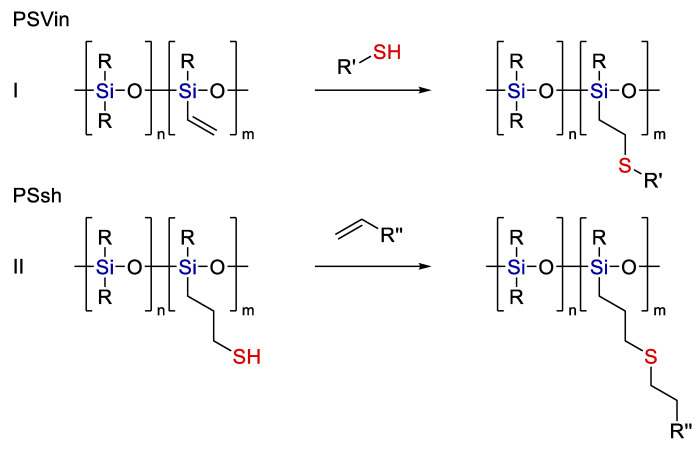
Variants for modification of polyorganosiloxane polymers by thiol-ene addition.

**Figure 25 polymers-14-03079-f025:**
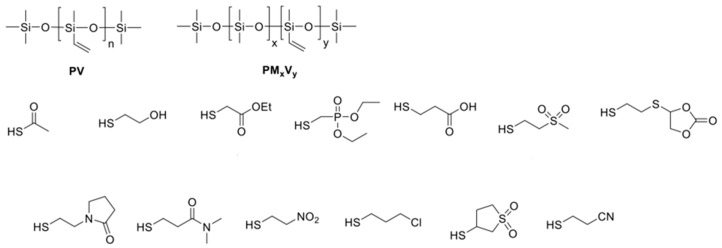
The modification of PSVin with thiols of various nature. Reprinted/adapted with permission from Ref. [103].

**Figure 26 polymers-14-03079-f026:**
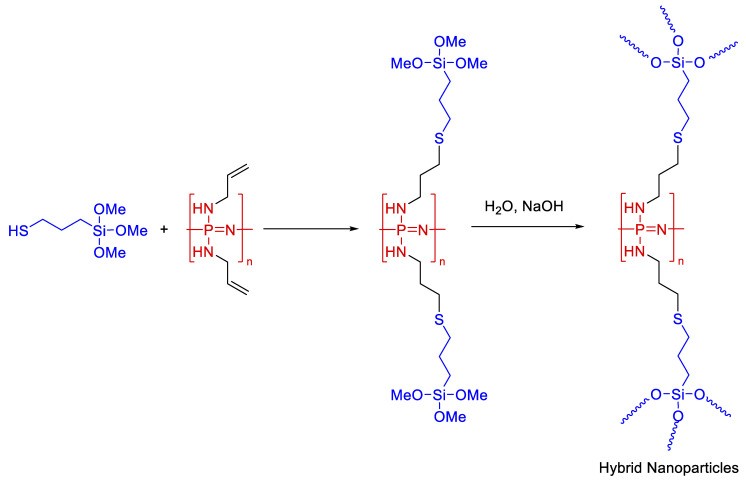
The synthetic procedure for the silica-based hybrid nanoparticles preparation using linear linear trimethoxysilane with phosphazene backbone precursor. Adopted from Ref. [114].

**Figure 27 polymers-14-03079-f027:**
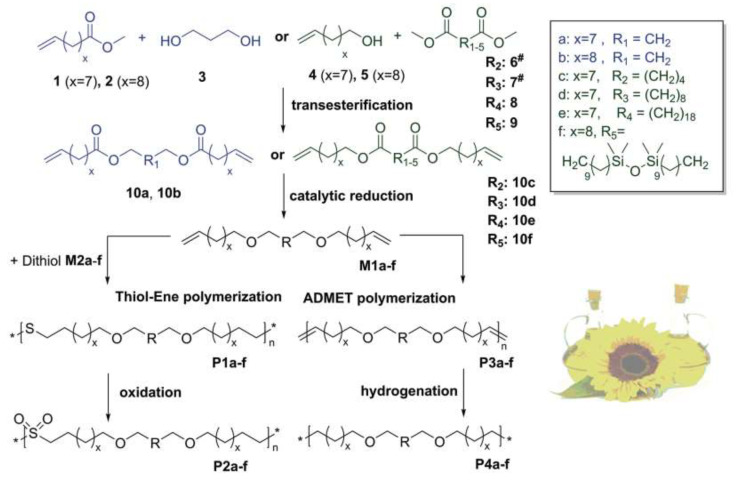
An example of performing the Ht and Hs reactions with natural substrates. Reproduced with permission from Ref. [125]. Copyright 2019 John Wiley and Sons.

**Figure 28 polymers-14-03079-f028:**
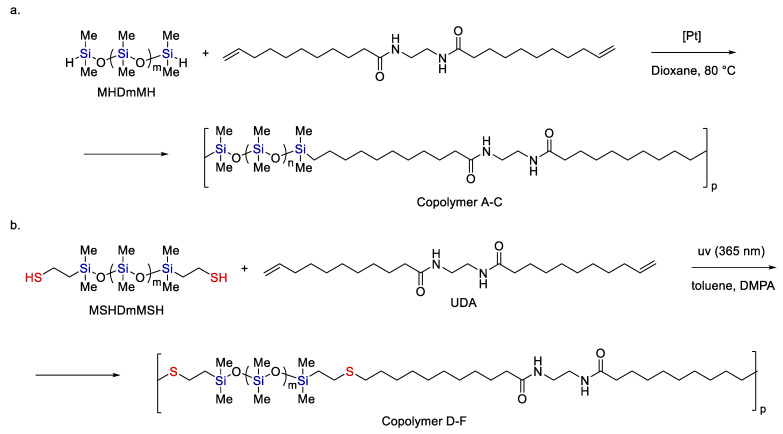
The synthesis of siloxane-containing copolymers based on undecenoic acid diamide by hydrosilylation (**a**) and hydrothiolation (**b**). Adopted from Ref. [126].

**Figure 29 polymers-14-03079-f029:**
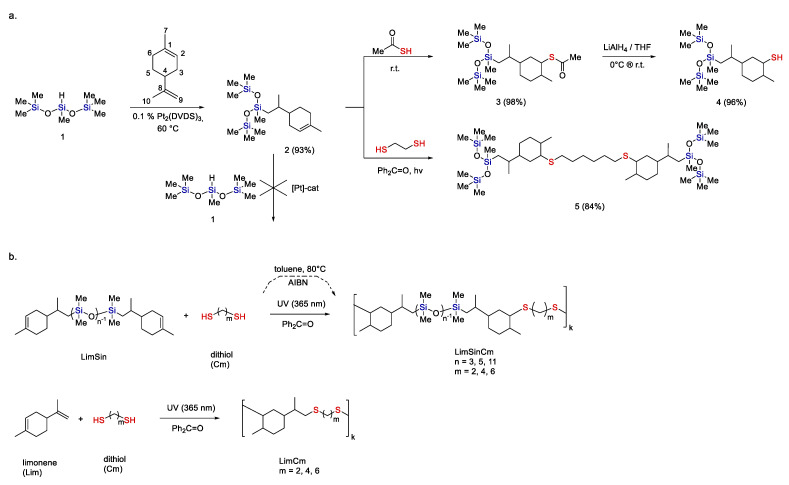
Strategies for the synthesis of functional limonene derivatives by hydrosilylation and hydrothiolation (**a**) [46]; synthesis of limonene-based copolymers by thiol-ene polyaddition (**b**) [127]. Adopted from Refs. [46,127].

**Figure 30 polymers-14-03079-f030:**
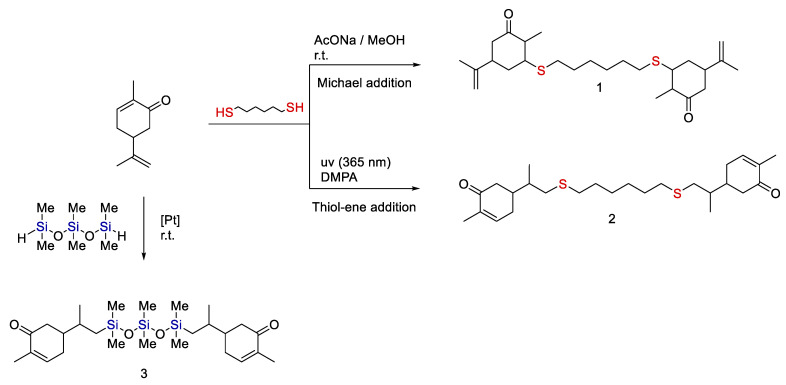
Strategies for the synthesis of functional derivatives of carvone by hydrosilylation and hydrothiolation. Adopted from Ref. [129].

**Figure 31 polymers-14-03079-f031:**
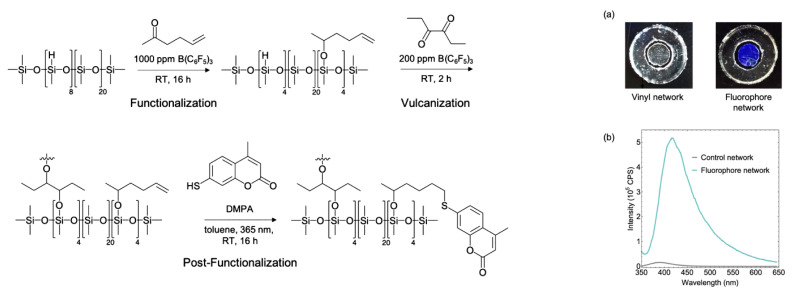
A scheme for synthesizing a cross-linked silicone by hydrosilylation involving a diketone followed by hydrothiolation with a thiol-containing coumarin dye. Reproduced with permission from Ref. [130]. Copyright 2019 American Chemical Society. (**a**) Images of the vinyl-functionalized network before (left) and after (right) thiol−ene click with 7-mercapto-4- ethylcoumarin under 365 nm irradiation. Center disk has undergone thiol−ene click, while the outer ring is an unreacted control sample. (**b**) Fluorescence spectra of the coumarin-containing network as well as the undecorated control.

**Figure 32 polymers-14-03079-f032:**
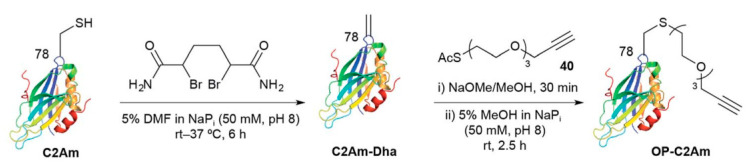
The application of a combination of Hs and Ht reactions for the addition of an organosilicon derivative of hippuric acid to synaptotagmin. Republished with permission of Royal Society of Chemistry, © 2017, from Ref. [131].

**Figure 33 polymers-14-03079-f033:**
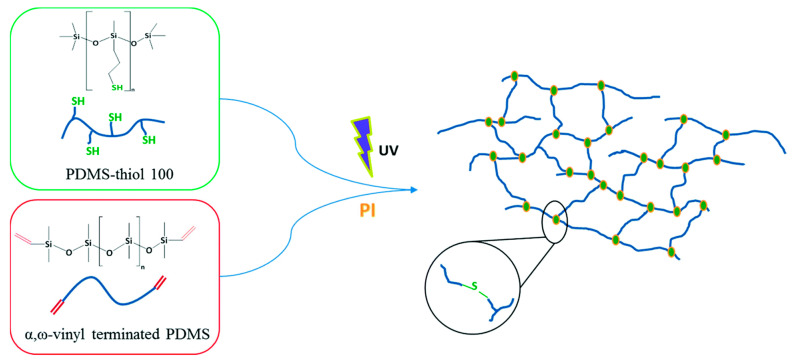
The synthesis of cross-linked polydimethylsiloxane. Republished with permission of Royal Society of Chemistry, © 2016, from Ref. [132].

**Figure 34 polymers-14-03079-f034:**
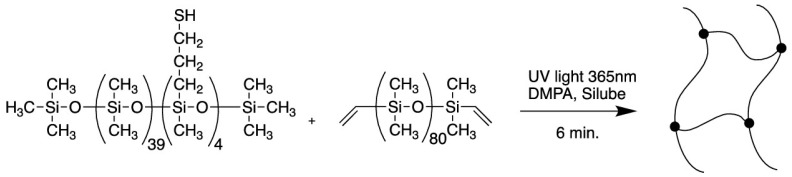
The synthesis of a cross-linked polydimethylsiloxane by hydrothiolation. Reproduced with permission from Ref. [135]. Copyright 2020 American Chemical Society.

**Figure 35 polymers-14-03079-f035:**
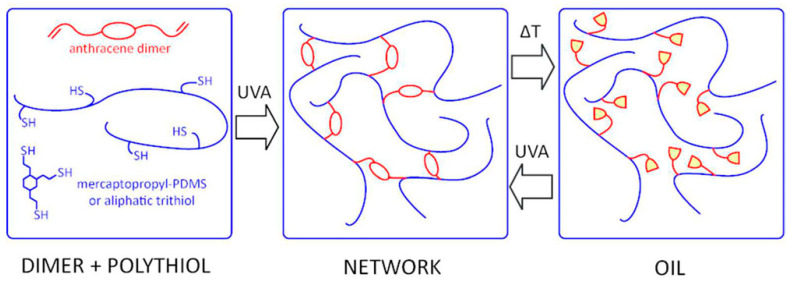
The cross-linked recyclable polymers. Reproduced with permission from Ref. [136]. Copyright 2017 American Chemical Society.

**Figure 36 polymers-14-03079-f036:**
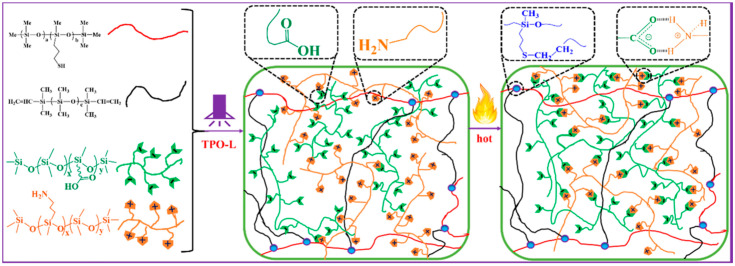
Self-healing polysiloxane elastomers. Reproduced with permission from Ref. [138]. Copyright (2020), with permission from Elsevier.

**Figure 37 polymers-14-03079-f037:**
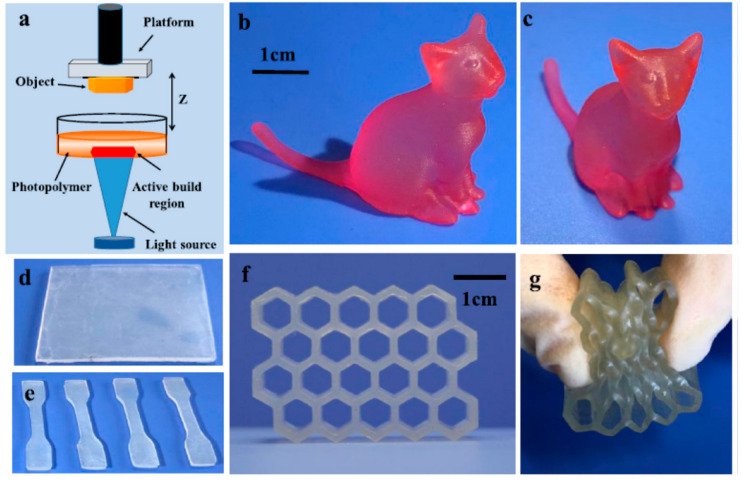
Elastomers based on cross-linked polydimethylsiloxane promising for application in 3D printing. Reproduced with permission from Ref. [139]. Copyright 2019 American Chemical Society. (**a**) Illustration of the working principle of DLP printing. (**b**,**c**) Photographs of the printed cat with the photopolymer resin L-0 (the red color is obtained by putting the printed cat in the red ink). (**d**,**e**) Photographs of the printed square samples for tensile test with the photopolymer resin HPsi-20 and the derived dumbbell-shaped test samples. (**f**,**g**) Photographs of the printed honeycomb-shaped sample with the photopolymer L-0.

**Figure 38 polymers-14-03079-f038:**
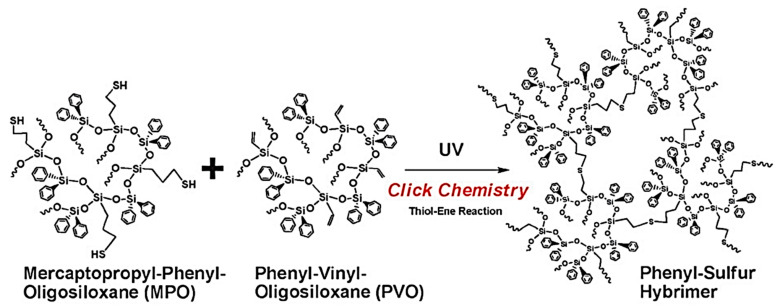
A new hybrid materials obtained by UV-initiated hydrothiolation. Republished with permission of Royal Society of Chemistry, © 2011, from Ref. [142].

**Figure 39 polymers-14-03079-f039:**
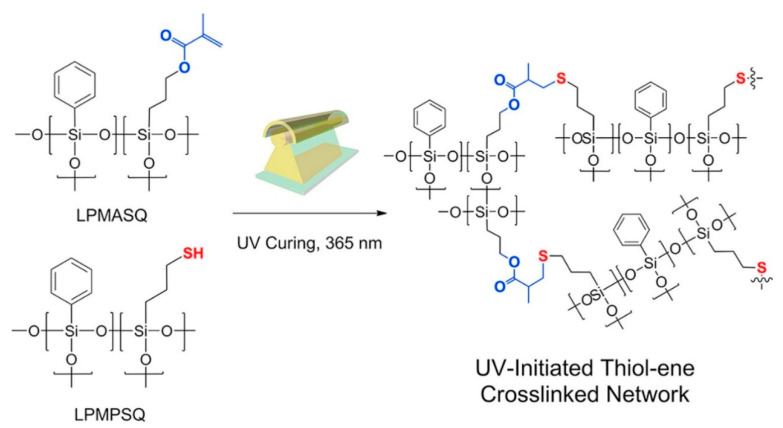
The synthesis of thermoplastic UV-curable polyphenylsilsesquioxanes. Reproduced with permission from Ref. Copyright (2015), with permission from Elsevier [144].

**Figure 40 polymers-14-03079-f040:**
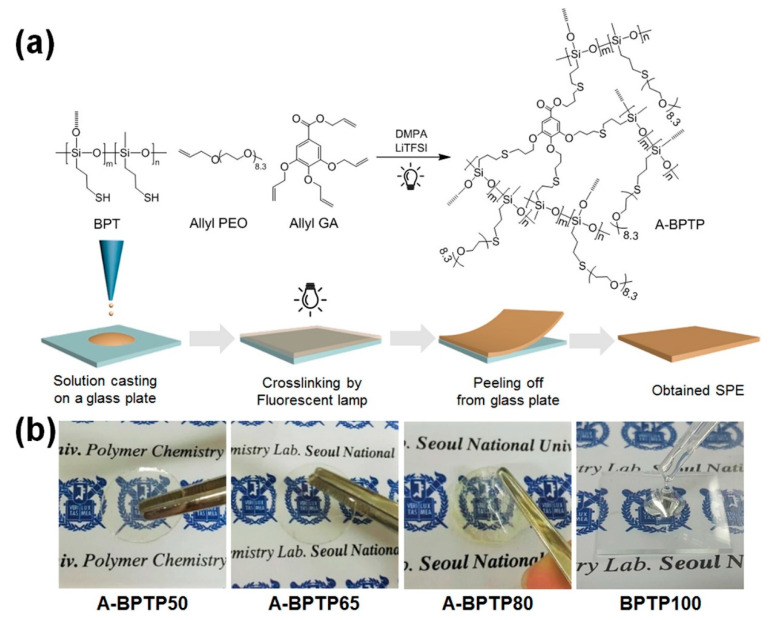
A material for lithium-ion batteries based on cross-linked polysiloxane. Reproduced with permission from Ref. [160]. Copyright (2017), with permission from Elsevier. (**a**) Preparation of A-BPTPs by thiol-ene click reaction under fluorescent lamp irradiation and (**b**) photographs of A-BPTPs having free-standing film states and BPTP100 having a wax state, where the number in the name indicates mol% of allyl PEO.

**Figure 41 polymers-14-03079-f041:**
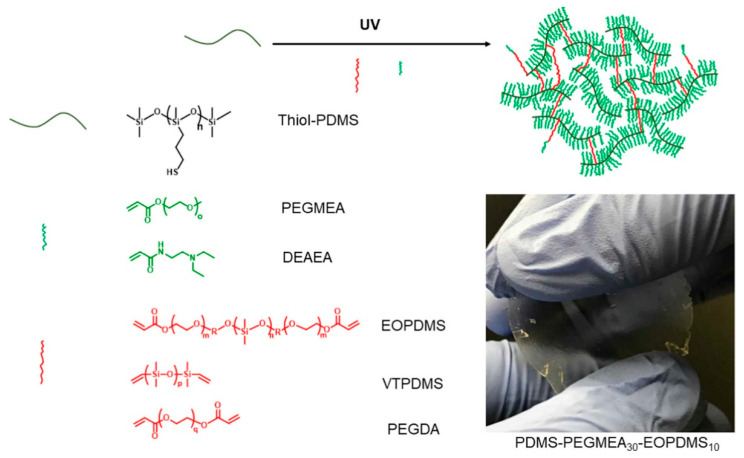
Preparation of elastomeric membranes. Reproduced with permission from Ref. [161]. Copyright 2019 American Chemical Society.

**Figure 42 polymers-14-03079-f042:**
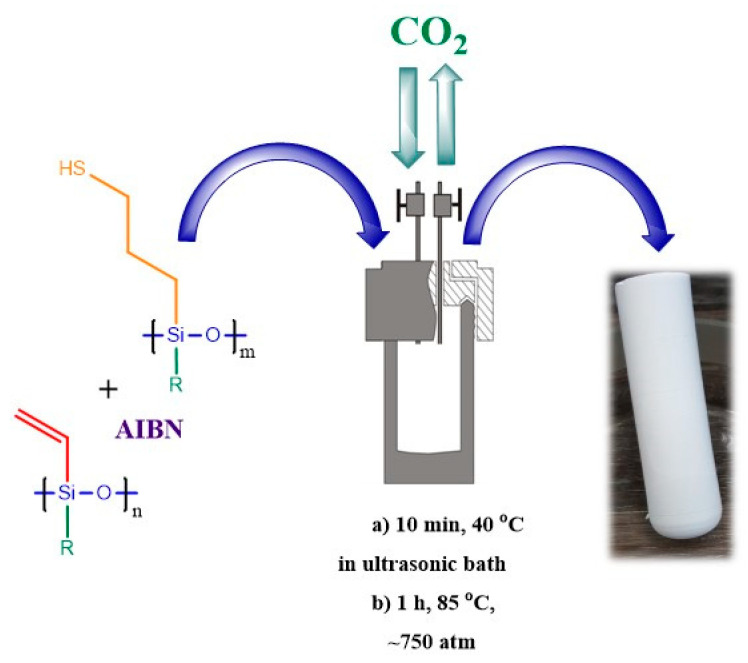
A schematic representation of aerogel preparation directly in sc-CO_2._ Reproduced with permission from Ref. [163]. Copyright (2018), with permission from Elsevier.

## Data Availability

Not applicable.

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
