# Peer review of "The Use of the Thiol-Ene Addition Click Reaction in the Chemistry of Organosilicon Compounds: An Alternative or a Supplement to the Classical Hydrosilylation?"

_polymers, 2022, doi:10.3390/polym14153079_

Round 1
Reviewer 1 Report
The Authors presented a very interesting review describing recent advances in the thiol addition chemistry with a particular emphasis on its application in organosilicon synthesis. The authors focused on the comparison of thiol addition to catalytic hydrosilylation and listed all the drawbacks and limitations originating from mentioned processes. The manuscript is well organized and written. Therefore, I recommend this review for publication after a very minor revision.
-Figure 2. The ratio between the alpha and beta products of triisopropylsilyl derivatives should be reported as it was done for triethylsilyl derivatives.
- page 9, it is worth mentioning that benzothiazole derivatives can be also prepared via hydrosilylation (DOI: 10.1016/j.jorganchem.2017.07.007)
- Page 9, line 294, The Authors postulate "Moreover, the combination of hydrothiolation with subsequent hydrosilylation looks quite promising, though it has yet not been studied in sufficient detail. It was shown convincingly [40] that upon hydrothiolation of silanes containing hydride at silicon, this Si-H bond is preserved and is theoretically suitable for subsequent hydrosilylation" but in the next paragraph, the Authors mentioned "It is worthy of note that, despite the demand for the hydrosilylation reaction in the synthesis of new POSS, it has a number of limitations. This reaction occurs very poorly or does not occur at all if the substrate contains heteroatoms (N, O, S, etc.)..." Therefore I would like to ask: Did the Authors find any papers describing the synthesis of novel organosilicon derivatives through thiol addition and subsequent hydrosilylation? Even if there are failed experiments, it should be also mentioned in the text. I am asking about that because the Authors mentioned in the text, that there is a problem with hydrosilylation of N, S-bearing compounds. Moreover, the combination of both processes but in the opposite order has been published by Marciniec et al. (DOI: 10.1039/C9NJ04488D), please add to the references.
- I would like to go back to the sentence about the limitations of hydrosilylation "It is worthy of note that, despite the demand for the hydrosilylation reaction in the synthesis of new POSS, it has a number of limitations. This reaction occurs very poorly or does not occur at all if the substrate contains heteroatoms (N, O, S, etc.) or groups with a mobile hydrogen atom (acids, alcohols, amines, etc.). If this is the case, protective groups have to be introduced or new catalytic systems have to be found" I think that this comment is more true to hydrosilylation of all C=C-containing compounds (vinylsilanes, olefins, vinylsiloxanes, silsesquioxanes) and not only limited to POSS derivatives. Therefore, I recommend a slight modification of this part. Moreover, it should be mentioned that significant progress in the hydrosilylation of S, N-containing alkenes has been achieved (for example: 10.1021/acs.joc.8b02838). However, the reported procedure requires a high Ir loading when compared to the typical Pt-hydrosilylation procedures.
- page 19, "In another example, an acrylonitrile-butadiene rubber was modified with a fluorosilicon rubber. The properties of blends of a modified (MNBR/FSR) and unmodified acrylonitrile-butadiene rubber with a fluorosilicon rubber (NBR/FSR) were compared" I think that the Authors should mention the significant differences in the hydrosilylation and hydrothiolation of synthetic (butadiene-based) rubbers. In light of the available scientific data, the selectivity of hydrothiolation of polybutadiene is very low. There are more side reactions than for monoalkenyl compounds (for example: 10.1021/ma802047w, 10.1021/ma100966w, etc.) On the other hand, the hydrosilylation of synthetic rubber is a highly selective and functional group tolerant process (DOI: 10.1021/ma052508y, 10.1016/j.jcat.2019.01.026 etc.). There are only two reports describing the dehydrogenative silylation of synthetic rubber, which is favored when the sterically hindered groups are directly bonded to a silicon atom (iPr- 10.1021/jm00073a007, tBu- 10.1016/j.jcat.2019.01.026)
- "Modification of polysiloxanes with natural compounds". The papers describing the synthesis and properties of eugenol-containing organosilicon compounds synthesized via HS and Ht should be added. I think that there are plenty of them.
Author Response
-Figure 2. The ratio between the alpha and beta products of triisopropylsilyl derivatives should be reported as it was done for triethylsilyl derivatives.
done
- page 9, it is worth mentioning that benzothiazole derivatives can be also prepared via hydrosilylation (DOI: 10.1016/j.jorganchem.2017.07.007)
done
- Page 9, line 294, The Authors postulate "Moreover, the combination of hydrothiolation with subsequent hydrosilylation looks quite promising, though it has yet not been studied in sufficient detail. It was shown convincingly [40] that upon hydrothiolation of silanes containing hydride at silicon, this Si-H bond is preserved and is theoretically suitable for subsequent hydrosilylation" but in the next paragraph, the Authors mentioned "It is worthy of note that, despite the demand for the hydrosilylation reaction in the synthesis of new POSS, it has a number of limitations. This reaction occurs very poorly or does not occur at all if the substrate contains heteroatoms (N, O, S, etc.)..." Therefore I would like to ask: Did the Authors find any papers describing the synthesis of novel organosilicon derivatives through thiol addition and subsequent hydrosilylation? Even if there are failed experiments, it should be also mentioned in the text. I am asking about that because the Authors mentioned in the text, that there is a problem with hydrosilylation of N, S-bearing compounds. Moreover, the combination of both processes but in the opposite order has been published by Marciniec et al. (DOI: 10.1039/C9NJ04488D), please add to the references.
- I would like to go back to the sentence about the limitations of hydrosilylation "It is worthy of note that, despite the demand for the hydrosilylation reaction in the synthesis of new POSS, it has a number of limitations. This reaction occurs very poorly or does not occur at all if the substrate contains heteroatoms (N, O, S, etc.) or groups with a mobile hydrogen atom (acids, alcohols, amines, etc.). If this is the case, protective groups have to be introduced or new catalytic systems have to be found" I think that this comment is more true to hydrosilylation of all C=C-containing compounds (vinylsilanes, olefins, vinylsiloxanes, silsesquioxanes) and not only limited to POSS derivatives. Therefore, I recommend a slight modification of this part. Moreover, it should be mentioned that significant progress in the hydrosilylation of S, N-containing alkenes has been achieved (for example: 10.1021/acs.joc.8b02838). However, the reported procedure requires a high Ir loading when compared to the typical Pt-hydrosilylation procedures.
done
- page 19, "In another example, an acrylonitrile-butadiene rubber was modified with a fluorosilicon rubber. The properties of blends of a modified (MNBR/FSR) and unmodified acrylonitrile-butadiene rubber with a fluorosilicon rubber (NBR/FSR) were compared" I think that the Authors should mention the significant differences in the hydrosilylation and hydrothiolation of synthetic (butadiene-based) rubbers. In light of the available scientific data, the selectivity of hydrothiolation of polybutadiene is very low. There are more side reactions than for monoalkenyl compounds (for example: 10.1021/ma802047w, 10.1021/ma100966w, etc.) On the other hand, the hydrosilylation of synthetic rubber is a highly selective and functional group tolerant process (DOI: 10.1021/ma052508y, 10.1016/j.jcat.2019.01.026 etc.). There are only two reports describing the dehydrogenative silylation of synthetic rubber, which is favored when the sterically hindered groups are directly bonded to a silicon atom (iPr- 10.1021/jm00073a007, tBu- 10.1016/j.jcat.2019.01.026)
Corrected
- "Modification of polysiloxanes with natural compounds". The papers describing the synthesis and properties of eugenol-containing organosilicon compounds synthesized via HS and Ht should be added. I think that there are plenty of them.
Chapter title changed
Reviewer 2 Report
Overall, this is a good review paper discussing using thiol-ene click reaction for organosilicon compounds, along with hydrosilylation reaction. This review paper did a good summary and discussion within the area and provided multi examples for each case. Some minor mistakes and/or error are found within the paper. And some minor revisions would be recommended. Please see the details below:
- I would recommend the authors tried to combine or reduces some of the figures within the paper, given currently 42 figures. Reducing the total number or the figures and combining some examples would be helpful.
- According to the original paper, alpha path with i-Pr is not possible. However, Fig 2 indicates both alpha and beta paths are possible when R=I-Pr
- Line 91, “It should be noted that there are few examples of this kind in the literature”, please provide the corresponding references.
- Figure 3 says “Figure 3. Scheme of the synthesis of carbosilane dendrimers by sequential Michael thiol-ene addition and radical thiol-ene addition reactions”, please indicate the Michael thiol-ene addition and radical thiol-one addition reactions within this scheme respectively.
- From line 107-112, the authors claimed several different variants for thiol-ene radical reaction. However, they seems not properly listed. For example, for line 109 “without an initiator” is a broader definition and includes line 112“ without a photo initiator”. The authors might want to differentiate thermal initiator and photo initiator
- In figure 6, the additional arrow with “polymers and copolymers” are very confusing. And it seems not very relevant with the main point of tis figure
- From line 275 to Lin 277, the authors claim “The number of alkoxy groups can vary from one to three, depending on the purpose. Examples of such structures and their applications are presented below. “ However, the following paragraph only discussed two alkoxy group examples.
- Figure 14 contains multi different examples within the figure. And the different functional groups for R, R’, R’’, R’’’, R’’’’ for each case are very difficult for the authors to understand it. I would suggest the authors try to improve the way of presentation
- In line 316, the authors suddenly use “Ht” (stands for hydrothiolation I guess) and Hs in line 325. Please make sure all the abbreviations are clearly defined before using.
- The caption for Fig. 26 is not clear. Please describe each portion of the figure clearly. For example, For the top right ones, it seems the element distribution within the nanoparticle. Is it for the surface or the bulk? And please make sure the captions for all figures are clear and easy to understand.
- Between line 549 and line 554, several abbreviations are used. Please make sure they are well defined.
- For section 5, the subtitle is “Modification of polysiloxanes with natural compounds”. However, for the examples presented in this section mainly discuss the different synthesis strategies/compatibility/selection of Ht and Hs reaction. Only Fig. 32 seems very relevant to the sub-title. I would recommend the authors to re-organize this section and/or change the sub-title to make it more relevant.
Author Response
- I would recommend the authors tried to combine or reduces some of the figures within the paper, given currently 42 figures. Reducing the total number or the figures and combining some examples would be helpful.
In our opinion, the amount of graphic material in the review is quite
corresponds to the required level of clarity and there are no extra pictures in the work.
2. According to the original paper, alpha path with i-Pr is not possible. However, Fig 2 indicates both alpha and beta paths are possible when R=I-Pr
done
3. Line 91, “It should be noted that there are few examples of this kind in the literature”, please provide the corresponding references.
Corrected
4. Figure 3 says “Figure 3. Scheme of the synthesis of carbosilane dendrimers by sequential Michael thiol-ene addition and radical thiol-ene addition reactions”, please indicate the Michael thiol-ene addition and radical thiol-one addition reactions within this scheme respectively.
done
5. From line 107-112, the authors claimed several different variants for thiol-ene radical reaction. However, they seems not properly listed. For example, for line 109 “without an initiator” is a broader definition and includes line 112“ without a photo initiator”. The authors might want to differentiate thermal initiator and photo initiator
done
6. In figure 6, the additional arrow with “polymers and copolymers” are very confusing. And it seems not very relevant with the main point of tis figure
Figure corrected
7. From line 275 to Lin 277, the authors claim “The number of alkoxy groups can vary from one to three, depending on the purpose. Examples of such structures and their applications are presented below. “ However, the following paragraph only discussed two alkoxy group examples.
In the articles to which we refer - a wide range of examples with different content of alkoxy groups. We considered it inappropriate to give examples of such a large number of structures. Moreover, additional examples (though more generally form) are shown in Figure 14.
8. Figure 14 contains multi different examples within the figure. And the different functional groups for R, R’, R’’, R’’’, R’’’’ for each case are very difficult for the authors to understand it. I would suggest the authors try to improve the way of presentation
Corrected
9. In line 316, the authors suddenly use “Ht” (stands for hydrothiolation I guess) and Hs in line 325. Please make sure all the abbreviations are clearly defined before using.
Corrected
10. The caption for Fig. 26 is not clear. Please describe each portion of the figure clearly. For example, For the top right ones, it seems the element distribution within the nanoparticle. Is it for the surface or the bulk? And please make sure the captions for all figures are clear and easy to understand.
Corrected
11. Between line 549 and line 554, several abbreviations are used. Please make sure they are well defined.
Corrected
12. For section 5, the subtitle is “Modification of polysiloxanes with natural compounds”. However, for the examples presented in this section mainly discuss the different synthesis strategies/compatibility/selection of Ht and Hs reaction. Only Fig. 32 seems very relevant to the sub-title. I would recommend the authors to re-organize this section and/or change the sub-title to make it more relevant.
Chapter title changed